# The research on small-scale structures of ice particle density and electron density in the mesopause region

Ruihuan Tian [1,2], Jian Wu[2], Jinxiu Ma[3], Yonggan Liang [1,2], Hui Li [1,2], Chengxun Yuan[1, 4], Yongyuan Jiang[1] and Zhongxiang Zhou[1, 4]

[1]Department of Physics, Harbin Institute of Technology, Harbin 150001, China

[2]National Key Laboratory of Electromagnetic Environment (LEME), China Research Institute of Radio Wave Propagation, Beijing 102206, China

[3]CAS Key Laboratory of Geospace Environment and Department of Modern Physics, University of Science and Technology of China, Hefei 230026, China

[4] Center of Space Environment of Polar Regions, Harbin Institute of Technology, Harbin 150001, China

Correspondence to: Hui Li (lihui_2253@163.com) and Chengxun Yuan (yuancx@hit.edu.cn)

**Abstract.** A growth and motion model of ice particles is originally developed based on the motion equation of a variable mass object to explain the formation of ice particle density irregularities with meter scale in the mesopause region. The growth of particles by adsorbing water vapor and the action of gravity and neutral drag force are considered in the growth and motion model. The evolution of radius, velocity, and number density of ice particles is investigated by solving the growth and motion model numerically. It is shown that, for certain nucleus radius, the velocity of particles can be reversed at particular height, which leads to local gathering of particles near the boundary layer. And then the small-scale ice particle density structures are formed. The spatial scale of the density structures can be affected by the vertical wind speed, water vapor density, and altitude. And these density structures can maintain stable as long as these environmental parameters do not change. The influence of these stable small-scale structures on electron and ion density is further calculated by a charging model, which considers the production, loss and transport of electrons and ions, and dynamic particle charging processes. The results show that, for particles with radii of 11 nm or less, the electron density is anti-correlated to the charged ice particle density and the ion density due to plasma attachment by particles and plasma diffusion, which is in accordance with most rocket observations. These small-scale electron density structures caused by small-scale ice particle density irregularities can produce the polar mesosphere summer echoes (PMSE) phenomenon.

# 1 Introduction

The polar mesosphere summer echoes (PMSE) are strong radar echoes from the polar mesopause in summer(Rapp and Lübken 2004). One of the features of PMSE is that the spectra widths of echoes are much narrower than that of incoherent scatter (being due to the Brownian movement of electrons)(Röttger, et al. 1988;Röttger, et al. 1990). And it has been proposed that the PMSEs are radar waves coherently scattered by the irregularities of the refractive index which are mainly determined by electron density(Rapp and Lübken 2004). Furthermore, the efficient scattering occurs when the spatial scale of electron density structures is half of the radar wavelength, the so-called Bragg scale. For typical VHF radars, the scale is about 3 m(Rapp and Lübken 2004). Experimentally, in the ECT02 campaign(Lübken, et al. 1998), the sounding rocket with electron probe has detected electron density irregularities on the order of meters during the simultaneous observation of PMSE, which provides a vital argument for that small-scale electron density structures can indeed create strong radar echoes.

Lots of researches indicate that small scale ice particle density irregularities in the PMSE region play a key role in creating and maintaining small-scale structures of electron density (Chen and Scales 2005;Lie‑Svendsen, et al. 2003;Mahmoudian and Scales 2013;Rapp and Lübken 2003;Scales and Ganguli 2004). Markus Rapp and Franz-Josef Lübken investigated electron diffusion in the vicinity of charged particles revisited (Rapp and Lübken 2003). They developed coupled diffusion equations for electrons, charged aerosol particles, and positive ions subject to the initial condition of anti-correlated perturbations in the charged aerosol and electron distribution. The results showed that the perturbations of electron density were anti-correlated to that of the negatively charged aerosol particles and positive ions. Ø. Lie-Svendsen et al studied the plasma response to imposed small-scale aerosol particle density perturbations (Lie‑Svendsen, et al. 2003). The results were consistent with the solution of Markus Rapp's model that particle density structures on the order of a few meters could lead to small-scale electron density perturbations due to electron attachment and ambipolar diffusion.

In all researches mentioned above, the aerosol particle density profiles were directly set as specific small scale structures such as Gaussian, hyperbolic tangent or sinusoidal. However the formation mechanism of these small-scale particle density structures has always been neglected, though they are helpful to understand PMSE phenomenon better. Kopnin et al. used dust acoustic solitons to explain the localized structures of the charged dust particles in the PMSE region (Kopnin, et al. 2004), but the

spatial scales of the obtained structures were much smaller than the observed scale and the wavelength of VHF radar. Therefore, it is still an open physical problem to study the formation mechanism of the small scale structures in PMSE region.

As is well-known, in the polar mesopause region, there is neutral airflow moving upward (Garcia and Solomon 1985). The ice particles are subjected to upward neutral drag force and downward gravity, and grow by absorbing water vapor simultaneously. In addition, the size of initial condensation nuclei has a certain distribution. These factors can cause complex trajectories of ice particles and result in an inhomogeneous distribution of particle number density, which then leads to small-scale structures of electron density. This may be an important mechanism that can produce PMSE phenomenon. But as far as we know, few people have studied the formation process of small-scale ice particle structures from the perspective of ice particle growth and movement.

In view of this, the particle growth and motion model is developed in this paper to describe the evolution of ice particle radius, velocity and density distribution in the mesopause region. The growth of particles is based on collision and adsorption process of condensation nuclei and water vapor. The particle movement is mainly controlled by the gravity and the neutral drag force. With the obtained ice particle density structures, the corresponding electron and ion densities are calculated based on a charging model, in which the continuity equations for ice particles with various charges and ions, the momentum equation for ions and electrons, and the quasi-neutral condition are included.

**2 Model**

In this section the equations of the growth and motion model of condensation nuclei and the charging model of ice particles are described.

The simulation is carried out at summer polar mesopause region between 80 ~ 90 km, where the water vapor carried by neutral gas is supposed to move upwards at a constant speed(Garcia and Solomon 1985). It is assumed that micrometeorites enter the study region at a certain flux from the upper boundary, and volcanic ash or particles ejected by aircraft rise into the region from the lower boundary. These grains serve as condensation cores. With the temperature lower than the frost point(Körner and Sonnemann 2001), the water vapor molecules that touch the surface of the grains due to thermal motion can easily condense into ice, which makes condensation cores become ice particles and keep growing. In this article, we will only discuss the growth, motion and charging process of

particles inside the condensation layer. Meantime, only vertical transport of particles and plasma is considered in this paper, because the horizontal gradients of transport parameters are much smaller than the vertical ones(Lie‐Svendsen, et al. 2003).

For growing ice particles, the dynamic equation for variable mass object is applied:

$$m_{\mathrm{d}} \frac{\mathrm{d}\boldsymbol{u}_{\mathrm{d}}}{\mathrm{d}t} + (\boldsymbol{u}_{\mathrm{d}} - \boldsymbol{u}) \frac{\mathrm{d}m_{\mathrm{d}}}{\mathrm{d}t} = m_{\mathrm{d}}\boldsymbol{g} - \mu_{\mathrm{dn}} m_{\mathrm{d}}(\boldsymbol{u}_{\mathrm{d}} - \boldsymbol{u}) + q_{\mathrm{d}}\boldsymbol{E} \qquad (1)$$

where $m_{\mathrm{d}}$, $\boldsymbol{u}_{\mathrm{d}}$ and $q_{\mathrm{d}}$ are the mass, velocity, and charge of ice particles respectively. $\boldsymbol{u}$ is the velocity of neutral gas; $\boldsymbol{g}$ is the gravitational acceleration; $\mu_{\mathrm{dn}}$ is the collision frequency between ice particles and gas; and $\boldsymbol{E}$ is the electric field. The electric force has trivial effect on the motion of ice particles, because

the charge-mass ratio of particles is usually very small(Jensen and Thomas 1988;Pfaff, et al. 2001). The inertial term is also negligible since its magnitude is much smaller than gravity (Garcia and Solomon 1985).

The water vapor is supersaturated in the polar mesopause region (Lübken 1999) and we assume that the size of condensation nuclei is larger than the condensation critical size, so stable growth of ice

particles will continue when water molecules collide with particles during thermal motion. Ignoring reverse process such as sublimation, the mass change rate of ice particles is

$$\frac{\mathrm{d}m_{\mathrm{d}}}{\mathrm{d}t} = \mu_{\mathrm{wd}} m_{\mathrm{w}} \qquad (2)$$

The collision frequency between water vapor and ice particles is $\mu_{\mathrm{wd}} = n_{\mathrm{w}}\pi r_{\mathrm{d}}^{2} v_{\mathrm{w}}$ based on the hard-sphere collision model (Lieberman and Lichtenberg 2005). $m_{\mathrm{w}}$, $n_{\mathrm{w}}$ and $v_{\mathrm{w}}$ are mass, number density

and thermal velocity of water molecules, respectively.

The collision frequency between air molecules and ice particles in the neutral drag force term is(Schunk 1977)

$$\mu_{\mathrm{dn}} = \frac{8}{3\sqrt{\pi}} \frac{n_{\mathrm{n}} m_{\mathrm{n}}}{m_{\mathrm{d}} + m_{\mathrm{n}}} \sqrt{\frac{2k_{\mathrm{B}} T_{\mathrm{g}}(m_{\mathrm{d}} + m_{\mathrm{n}})}{m_{\mathrm{d}} m_{\mathrm{n}}}} \pi(r_{\mathrm{d}} + r_{\mathrm{n}})^{2} \qquad (3)$$

where $n_{\mathrm{n}}$, $m_{\mathrm{n}}$, and $r_{\mathrm{n}}$ are number density, mean molecule mass, and effective radius of neutral molecule,

respectively. $T_{\mathrm{g}}$ is the gas temperature. The neutral molecule mass $m_{\mathrm{n}}$ is assumed as $28.96 m_{\mathrm{u}}$. $m_{\mathrm{u}}$ is the proton mass.

From Eq. (1) we can get the velocity of ice particles

$$u_{\mathrm{d}} = u + \frac{m_{\mathrm{d}}}{\mu_{\mathrm{dn}} m_{\mathrm{d}} + \mu_{\mathrm{wd}} m_{\mathrm{w}}} g \qquad (4)$$

With the facts that $n_{\mathrm{w}} \ll n_{\mathrm{n}}$(Seele and Hartogh 1999), $m_{\mathrm{w}} \ll m_{\mathrm{d}}$, $m_{\mathrm{n}} \ll m_{\mathrm{d}}$, $r_{\mathrm{n}} \ll r_{\mathrm{d}}$ and $v_{\mathrm{n}} \sim v_{\mathrm{w}}$, and taking vertical up to be the positive direction, the velocity of ice particles is simplified as

$$u_{\mathrm{d}} = u - g/\mu_{\mathrm{dn}} \qquad (5)$$

Ice particles are composed of condensation nuclei and the attached ice. The mass of a single ice particle is

$$m_{\mathrm{d}} = \frac{4}{3} \pi r_0^3 \rho_0 + \frac{4}{3} \pi (r_{\mathrm{d}}^3 - r_0^3) \rho_{\mathrm{d}} \qquad (6)$$

where $r_0$ and $\rho_0$ are the initial radius and mass density of condensation nuclei, and $\rho_{\mathrm{d}}$ is the mass density of ice.

Based on the expressions of $m_{\mathrm{d}}$ and $\mu_{\mathrm{dn}}$, the relation between ice particle velocity and radius is

$$u_{\mathrm{d}} = u - \frac{g}{n_{\mathrm{n}} m_{\mathrm{n}} v_{\mathrm{n}}} [\rho_{\mathrm{d}} r_{\mathrm{d}} + (\rho_0 - \rho_{\mathrm{d}}) \frac{r_0^3}{r_{\mathrm{d}}^2}] \qquad (7)$$

At the boundaries of the condensation region, $r_{\mathrm{d}} = r_0$, and the initial velocity of condensation nuclei is

$$u_{\mathrm{d}0} = u(1 - r_0/r_{\mathrm{c}}) \qquad (8)$$

where $r_{\mathrm{c}}$ is the critical radius

$$r_{\mathrm{c}} = n_{\mathrm{n}} m_{\mathrm{n}} v_{\mathrm{n}} u/(g \rho_0) \qquad (9)$$

When the radius of condensation nuclei $r_0 > r_{\mathrm{c}}$, gravity is larger than the neutral drag force, $v_{\mathrm{d}0} < 0$, and particles move downwards. Otherwise, particles move upwards.

Based on the relation between $m_{\mathrm{d}}$ and $r_{\mathrm{d}}$, the change rate of ice particle radius is

$$\frac{\mathrm{d}r_{\mathrm{d}}}{\mathrm{d}t} = \frac{1}{4} \frac{n_{\mathrm{w}} m_{\mathrm{w}} v_{\mathrm{w}}}{\rho_{\mathrm{d}}} = c \qquad (10)$$

It is easy to see that the ice particle radius increases linearly with time

$$r_{\mathrm{d}} = r_0 + ct \qquad (11)$$

Then the particle trajectory can be obtained by the following integral

$$z - z_0 = \int_0^t u_d \mathrm{d}t = c^{-1} \int_{r_0}^{r_d} u_d \mathrm{d}r_d \tag{12}$$

$z_0$ is the reference height where condensation nuclei enter the studied region. It is set that $z_0 = 0$ at the lower boundary and $z_0 = h$ at the upper boundary, where $h$ is the distance between the two boundaries.

We assume that the condensation nucleus radius ranging from $r_{0\min}$ to $r_{0\max}$ has a certain distribution

function $f(r_0)$. The density of condensation nuclei with radius in the small range $r_0 \rightarrow r_0 + \mathrm{d}r_0$ is $\mathrm{d}n(r_0)$

$= f(r_0)\mathrm{d}r_0$, and their velocity is $u_{d0}$. When these particles arrive at height $z$, their radius increases to $r_d(r_0, z)$, the corresponding number density turns into $\mathrm{d}n(r_0, z)$, and the velocity becomes $u_d(r_0, z) = u_d[r_0, r_d(r_0, z)]$. According to the particle-conservation law, we have

$$u_{d0} \mathrm{d}n(r_0) = u_d(r_0, z) \mathrm{d}n(r_0, z) \tag{13}$$

Then the number density of ice particles at height $z$ can be obtained by

$$n_d(z) = \int \mathrm{d}n(r_0, z) = \int_{r_{0\min}}^{r_{0\max}} \frac{u_{d0} f(r_0)}{u_d(r_0, z)} \mathrm{d}r_0 \tag{14}$$

The averaged ice particle radius at height $z$ is

$$\overline{r_d}(z) = \frac{\int r_d(z) \mathrm{d}n(r_0, z)}{n_d(z)} \tag{15}$$

Through integrating all the condensation nucleus radii, stable distribution of $n_d$ and $r_d$ can be obtained. The particles keep entering and leaving the condensation region, and as long as the external

environment does not change, the distribution of particle density and radius will remain unchanged. Then the influence of these stable $n_d$ and $r_d$ profiles on electron and ion density is calculated.

Considering ionization, electron-ion recombination, and ion loss on ice particles, the continuity equation of ion density can be written as follow

$$\frac{\partial n_i}{\partial t} + \frac{\partial (n_i u_i)}{\partial z} = Q - \alpha n_i n_e - D^+ n_i \tag{16}$$

Ignoring gravity, the drift velocity of ions $u_i$ is determined by

$$u_i = \frac{eE}{m_i \mu_{in}} - \frac{k_B T_g}{m_i \mu_{in}} \frac{1}{n_i} \frac{\partial n_i}{\partial z} \tag{17}$$

The electric field $E$ is mainly determined by electron density gradient because the diffusion coefficient and mobility of electrons are much larger than that of ions:

$$E = -\frac{k_B T_g}{e} \frac{1}{n_e} \frac{\partial n_e}{\partial z} \tag{18}$$

In the typical PMSE layer, there are several kinds of ions carrying one unit positive charge: $N_2^+$, $O_2^+$, $NO^+$ and $H^+(H_2O)_n$. According to Ref. (Reid 1990), the averaged ion parameters $n_i$, $m_i$, and $T_g$ are applied to describe the density, mass, and temperature of ions, respectively, and the averaged ion mass $m_i$ is set as $50m_u$ at 85 km altitude. According to Hill and Bowhill's theory (Hill and Bowhill 1977), the ion-neutral collision frequency is

$$\mu_{in} = 2.6 \times 10^{-15} n_n \left( 0.78 \frac{28}{M_i + 28} \sqrt{1.74 \frac{M_i + 28}{28 M_i}} \right.$$
$$\left. + 0.21 \frac{32}{M_i + 32} \sqrt{1.57 \frac{M_i + 32}{32 M_i}} + 0.01 \frac{40}{M_i + 40} \sqrt{1.64 \frac{M_i + 40}{40 M_i}} \right) \tag{19}$$

where $M_i = m_i/m_u$.

The production rate for ions and electrons $Q$ is chosen as $3.6 \times 10^7$ m$^{-3}$s$^{-1}$ and electron-ion recombination coefficient $\alpha$ is set as $10^{-12}$ m$^3$s$^{-1}$ (Lie‐Svendsen, et al. 2003). Then the undisturbed density of ions and electrons $n_0 = 6 \times 10^9$ m$^{-3}$. The loss coefficient of ions on ice particles $D^+ = \Sigma n_q v_{i,q}$, where $n_q$ is the number density of the $q$-charged ice particles, and $v_{i,q}$ represents the capture rate of ions by ice particles with $q$ charges. According to the discrete charging model (Robertson and Sternovsky 2008):

$$v_{i,q \le 0} = \pi r_d^2 c_i \left( 1 + C_q \sqrt{\frac{e^2}{16 \varepsilon_0 k_B T_g r_d}} + D_q \frac{e^2}{4 \pi \varepsilon_0 k_B T_g r_d} \right) \tag{20}$$

The particle radius $r_d$ used here is the averaged radius $\bar{r}_d$, which is obtained according to Eq. (15). The ion thermal velocity $c_i = (8k_B T_g/\pi m_i)$. $k_B$ is Boltzmann's constant and $\varepsilon_0$ is the permittivity of vacuum. $C_q$ and $D_q$ are given in Table 1 of Robertson and Sternovsky's work (Robertson and Sternovsky 2008). And the corresponding capture rates of electrons by ice particles (Robertson and Sternovsky 2008) are written as

$$v_{e,q \ge 0} = \pi r_d^2 c_e \left( 1 + C_q \sqrt{\frac{e^2}{16 \varepsilon_0 k_B T_g r_d}} + D_q \frac{e^2}{4 \pi \varepsilon_0 k_B T_g r_d} \right) \tag{21}$$

$$v_{\mathrm{e},q<0} = \pi r_{\mathrm{d}}^2 \gamma^2 c_{\mathrm{e}} \exp\left[ -\frac{|q|e^2}{4\pi\varepsilon_0 k_{\mathrm{B}} T_{\mathrm{g}} r_{\mathrm{d}}\gamma}\left(1 - \frac{1}{2\gamma(\gamma^2-1)|q|}\right)\right] \tag{22}$$

The thermal velocity of electrons $c_{\mathrm{e}} = (8k_{\mathrm{B}}T_{\mathrm{g}}/\pi m_{\mathrm{e}})$, and the value of $\gamma$ for each $q$ is referred from Natanson's paper (Natanson 1960).

Although the distribution of total particle density $n_{\mathrm{d}} = \Sigma n_q$ has reached stable state under the action of gravity and neutral drag force, the number density of the $q$-charged ice particles $n_q$ is dynamic in the charging process. The continuity equation of $q$-charged ice particle density is

$$\frac{\partial n_q}{\partial t} = n_{q+1}v_{\mathrm{e},q+1}n_{\mathrm{e}} + n_{q-1}v_{\mathrm{i},q-1}n_{\mathrm{i}} - (n_q v_{\mathrm{e},q}n_{\mathrm{e}} + n_q v_{\mathrm{i},q}n_{\mathrm{i}}) \tag{23}$$

According to the work in references (Lie‐Svendsen, et al. 2003;Rapp and Lübken 2001), it is assumed that a single particle carries two negative charges at most, i.e., $q$ = -2, -1, 0 and +1 in this study.

According to the typical parameters in PMSE region(Rapp and Lübken 2001), the plasma Debye length $\lambda_{\mathrm{D}}$ is estimated to be about 9 mm, which is much smaller than the vertical spatial scale of PMSE layer. So the dusty plasma satisfies the quasi-neutral condition:

$$n_{\mathrm{i}} + \sum_q q n_q = n_{\mathrm{e}} \tag{24}$$

For simplicity, dimensionless parameters will be used in subsequent discussion:

$$V_{\mathrm{d}} = v_{\mathrm{d}}/u, \quad \rho = \rho_{\mathrm{d}}/\rho_0, \quad R_0 = r_0/r_{\mathrm{c}}, \quad R_{\mathrm{d}} = r_{\mathrm{d}}/r_{\mathrm{c}}$$
$$T = t/t_{\mathrm{c}}, \quad Z = (z - z_0)/z_{\mathrm{c}}$$

where $t_{\mathrm{c}} = r_{\mathrm{c}}/c$, which represents the time it takes for ice particles growing from $r_{\mathrm{d}}$ to $r_{\mathrm{d}} + r_{\mathrm{c}}$, and $z_{\mathrm{c}} = ut_{\mathrm{c}}$ is the distance that neutral wind moves during the time $t_{\mathrm{c}}$.

The expression of dimensionless ice particle velocity is

$$V_{\mathrm{d}} = 1 - \rho R_{\mathrm{d}} - (1-\rho)\frac{R_0^3}{R_{\mathrm{d}}^2} \tag{25}$$

The expressions of dimensionless position coordinate of particles based on $T$ and $R_{\mathrm{d}}$ are

$$Z(R_0, T) = T - \frac{1}{2}\rho T(T + 2R_0) - (1-\rho)R_0^2\frac{T}{T+R_0} \tag{26}$$

$$Z(R_0, R_{\mathrm{d}}) = R_{\mathrm{d}} - R_0 - \frac{1}{2}\rho(R_{\mathrm{d}}^2 - R_0^2) + (1-\rho)R_0^3(\frac{1}{R_{\mathrm{d}}} - \frac{1}{R_0}) \tag{27}$$

The dimensionless number density and radius distribution of ice particles are

$$n_d(Z) = n_0 \int_{R_{0min}}^{R_{0max}} \frac{V_{d0}F(R_0)}{V_d[R_0, R_d(R_0, Z)]} dR_0 \tag{28}$$

$$\bar{R}_d(Z) = \frac{n_0}{n_d(Z)} \int_{R_{0min}}^{R_{0max}} \frac{R_d(Z)V_{d0}F(R_0)}{V_d[R_0, R_d(R_0, Z)]} dR_0 \tag{29}$$

where $n_0$ is the density of condensation cores at the boundary, and is assumed as $5 \times 10^8$ m$^{-3}$ (Bardeen, et al. 2008). The normalized radius distribution function $F(R_0)$ satisfies $\int_{R_{0min}}^{R_{0max}} F(R_0)dR_0 = 1$.

In subsequent calculations, parameters are taken in the atmospheric environment at altitude of 85 km. The number density of neutrals $n_n = 2.3 \times 10^{20}$ m$^{-3}$(Hill, et al. 1999), the number density of water vapor $n_w = 2.5 \times 10^{14}$ m$^{-3}$(Seele and Hartogh 1999), temperature $T_g = 150$ K, the mass density of ice $\rho_d = 1 \times 10^3$ kg/m$^3$, the velocity of neutral wind $u = 3$ cm/s(Garcia and Solomon 1985), the mass density of condensation nucleus $\rho_0 = 2.7 \times 10^3$ kg/m$^3$, and the growth rate of ice particles $c \approx 7.8 \times 10^{-4}$ nm/s. In this work, we only consider the growth and movement of condensation nuclei which fall from the upper boundary with initial radius $r_0 > r_c$ and rise from lower boundary with $r_0 \le r_c$.

## 3 Results and discussion

### 3.1 The speed and trajectory of ice particles

The relation between $V_d$ and $R_d$ is illustrated in Fig. 1(a), which shows that condensation nuclei with initial radius $R_0 \le 1$ rise into the PMSE region through the lower boundary, while particles with $R_0 > 1$ fall into the region from the upper boundary. At the beginning, the upward-moving particles accelerate and the downward ones decelerate due to $\partial V_d/\partial R_d = 2 - 3\rho > 0$ when $R_d = R_0$. Later, with the increase of $R_d$, $\partial V_d/\partial R_d < 0$, all particles will move with a downward acceleration, which makes them move downward eventually.

Figure 1(b) shows the movement curves of ice particles near the lower boundary. These particles, with an initial radius $R_0 \le 1$, rise into the condensation layer. With the collection of ice, the grains become larger and heavier, which leads to the deceleration of the grains. And then, the grains will accelerate downward until they leave the condensation layer from the lower boundary. All particles rising from the lower boundary will retrace in the range $Z_m < Z < Z_M$.

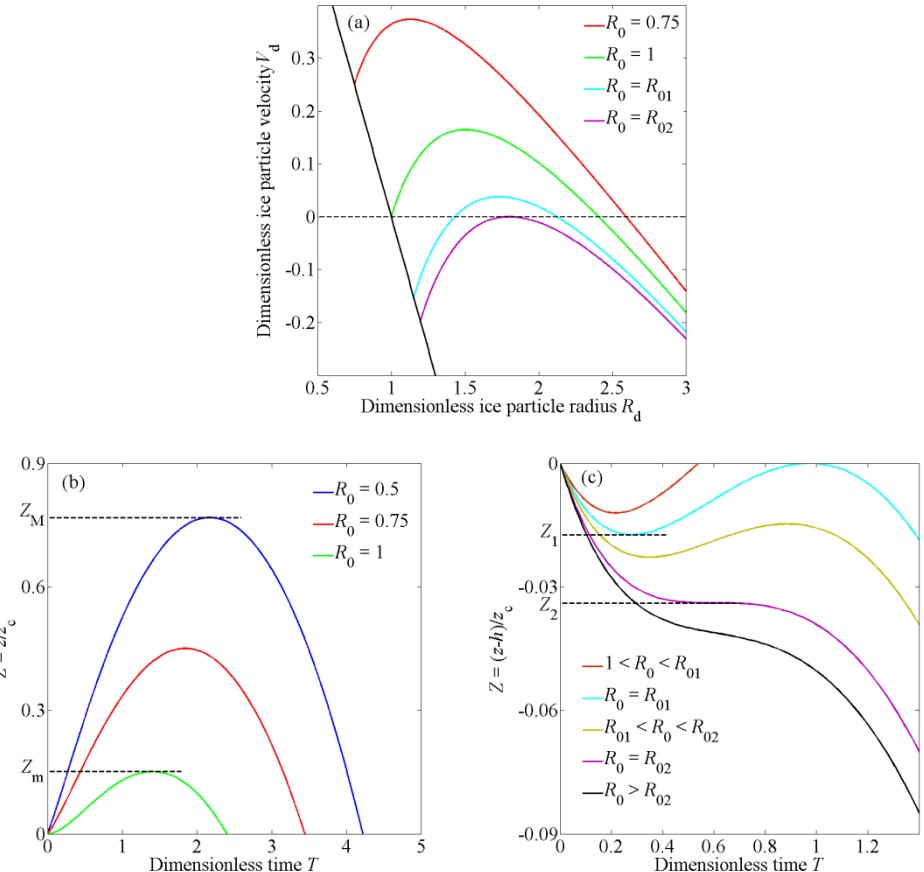

Figure 1 (a) The dependence of the ice particle velocity on radius for different initial nucleus radii. The black solid line $V_{d0} = 1 - R_0$ represents the relation between the initial particle velocity and the initial particle radius. (b) The movement curves of ice particles near the lower boundary. (c) The movement curves of ice particles near the upper boundary. $Z_m$ is the maximum height that particles with initial radius $R_0 = 1$ can reach; $Z_M$ is the maximum height that particles with initial radius $R_0 = R_{0min} = 0.5$ can reach. Based on above parameters, $Z_m = 0.1512$ and $Z_M = 0.7682$. $R_{01}$ and $R_{02}$ are two critical values of condensation nucleus radius. For $R_0 = R_{01}$, particles fall into the condensation layer, first retrace at height $Z_1$, and then retrace exactly at the upper boundary. When $R_0 = R_{02}$, the particles move down and reach the height $Z_2$ with the velocity and acceleration being exactly zero, and then they continue to move down. According to above parameters, $R_{01}$ and $R_{02}$ are solved as 1.1519 and 1.19705, respectively.

Figure 1(c) shows the movement curves of ice particles near the upper boundary, which can be sorted by the value of $R_0$. For $1 < R_0 < R_{01}$, the neutral drag force increases faster than gravity as the particles

fall. The particles decelerate to zero speed, retrace upward, and then leave the condensation layer from the upper boundary. For $R_0 = R_{01}$, the particles retrace at the height $Z = Z_1$. Then they arrive at $Z = 0$ with exactly zero velocity, and the particles move back into the condensation layer again. For $R_{01} < R_0 < R_{02}$, the particles retrace upward in the range of $Z_2 < Z < Z_1$ and move downward again before they reach the upper boundary. For $R_0 = R_{02}$, the particles decelerate downward until zero speed at $Z = Z_2$. Here, the acceleration happens to be zero. Then the gravity exceeds the drag force, and the particles accelerate downward. For $R_0 > R_{02}$, the particles keep going down after entering the condensation layer.

From Fig. 1, it can be seen that the particles with certain initial radius will move up and down several times near the boundary, namely, ice particles will accumulate at that region and form some kind of small-scale density structure.

## 3.2 The density and radius distribution of ice particles and their effects on plasma

### 3.2.1 Near the lower boundary

Firstly, the density and radius distribution of ice particles near the lower boundary are solved. It is shown in Fig. 1(b) that all ice particles with initial radius $R_0 \leq 1$ will pass the range $0 < Z < Z_m$ twice, so they contribute twice to the calculation of particle density. And in the height range $Z_m < Z < Z_M$, only the particles that can reach the $Z$ height will contribute to the density at $Z$. The density and mean radius of ice particles near the lower boundary are shown below:

$$n_d(Z) = n_0 \int_{0.5}^{R_{0Z}} V_{d0} F(R_0) \left[ \frac{1}{V_{d1}(R_0, R_{d1})} + \frac{1}{|V_{d2}(R_0, R_{d2})|} \right] dR_0 \qquad (30)$$

$$\bar{R}_d(Z) = \frac{n_0}{n_d(Z)} \int_{0.5}^{R_{0Z}} V_{d0} F(R_0) \left[ \frac{R_{d1}}{V_{d1}(R_0, R_{d1})} + \frac{R_{d2}}{|V_{d2}(R_0, R_{d2})|} \right] dR_0 \qquad (31)$$

$R_{d1}$ and $R_{d2}$ are particle radii when particles pass through the $Z$ height; $V_{d1}$ and $V_{d2}$ are the corresponding velocities; the upper limit of integral $R_{0Z}$ is determined by

$$R_{0Z} = \begin{cases} 1 & \text{if } 0 < Z < Z_m \\ \text{solution of } (Z(R_{0Z}, R_d) = Z) & \text{if } Z_m < Z < Z_M \end{cases} \qquad (32)$$

In this study, the radius distribution function of condensation cores is assumed as Gaussian distribution

$$F(R_0) = A\exp[-(R_0 - R_{00})^2 / \Delta^2] \qquad (33)$$

where the center of the radius distribution function $R_{00}$ is chosen as 0.8, the characteristic width $\Delta = $ 0.01, and the corresponding normalized coefficient A = 56.4.

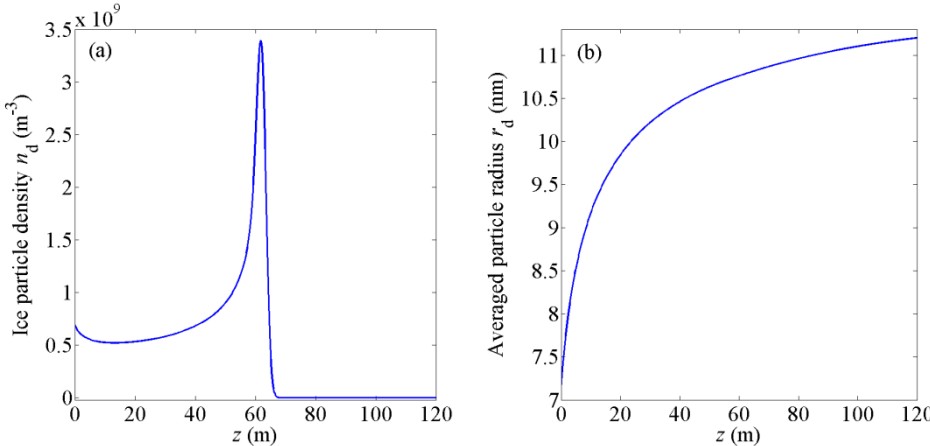

Figure 2 The distribution of (a) ice particle density and (b) the averaged particle radius near the lower boundary of the condensation layer.

The obtained density and mean radius of ice particles near the lower boundary are presented in Fig. 2(a) and 2(b) respectively. Figure 2(a) shows that a sharp peak appears in the density distribution of ice particles. The width at half maximum of the irregularity is about 5 meters, which is consistent with the assumed ice particle density structure scale in the theoretical work (Lie-Svendsen, et al. 2003;Rapp and Lübken 2003) and observation by the sounding rocket flight ECT02 in July 1994 (Rapp and Lübken
2004). From Fig. 2(b), we can see that the average radius of ice particles increases from 7 nm to 11 nm with height.

  With the obtained density and average radius of ice particles in Fig. 2(a) and Fig. 2(b), the density distribution of electrons, ions, and charged ice particles is calculated based on the charging model described by Eq. (16) ~ (24). At the initial moment of the charging model, all ice particles are assumed
to be neutral to conduct the calculation more conveniently, since the final distributions of charge are independent on the initial ice particle charge state (Lie‐Svendsen, et al. 2003). The timescale of electron collected by negatively charged particles with a radius of 10 nm is about 700 s, which is the longest timescale in the charging process. And a quasi-steady state of charging can be obtained after this timescale. Therefore, the calculation is terminated after 1000 s and the results are illustrated in Fig. 3.

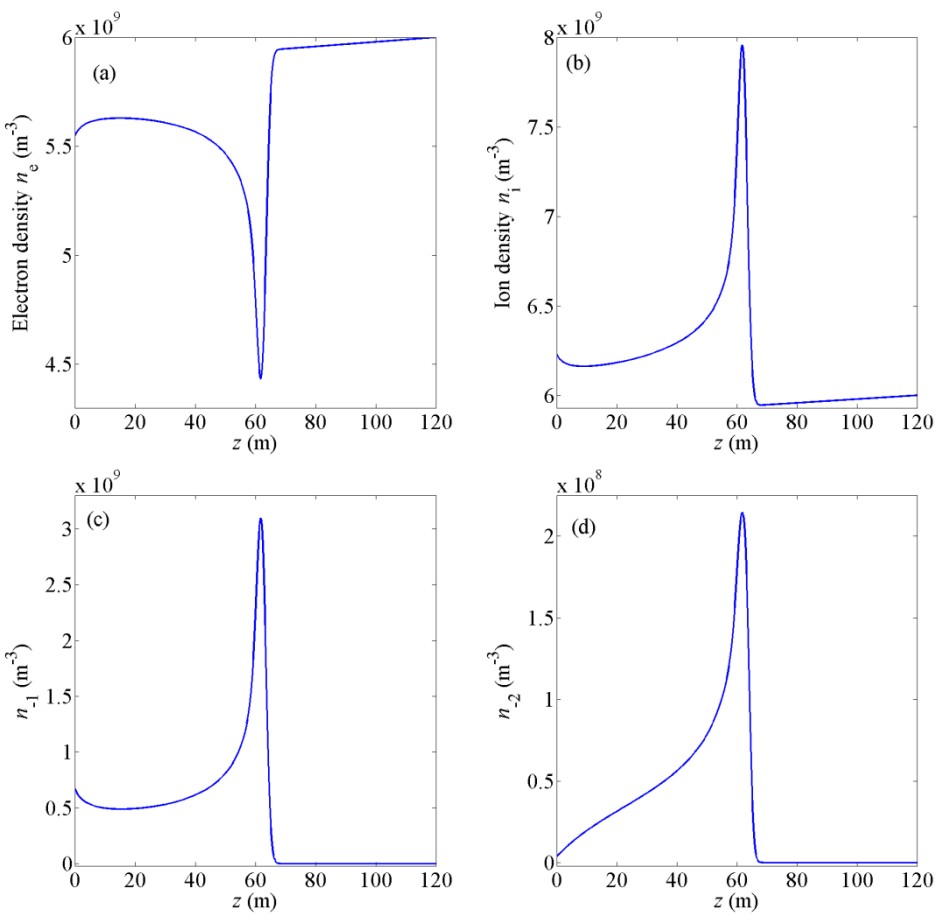

Figure 3 The number density distribution of (a) electrons $n_e$, (b) ions $n_i$, (c) particles carrying one negative charge $n_{-1}$, and (d) particles carrying two negative charges $n_{-2}$ near the lower boundary of condensation layer at $t = 1000$ s.

Figure 3(a) shows that electron density decreases sharply around $z = 60$ m due to adsorption by particles. And the reduction of electron density $\Delta n_e \approx (n_{-1} + 2n_{-2})/2$, which is in line with the results under diffusion equilibrium approximations in (Lie‑Svendsen, et al. 2003). Ion number density increases sharply around 60 m due to the diffusion under ambipolar electric field. The ambipolar diffusion process of electrons and ions has been described in detail in (Lie‑Svendsen, et al. 2003). Electron density is anti-correlated to density irregularities of ions and the charged ice particles due to attachment and diffusion processes. These anti-correlations are in agreement with rocket observations by the sounding rocket flight SCT-06 in August 1993 (Lie‑Svendsen, et al. 2003) and the sounding rocket flight ECT02 in July 1994 (Rapp and Lübken 2004), respectively. It can be extracted from Fig.

3(c) and Fig. 3(d) that, for particles with radii ranging from 7 nm to 11 nm, the proportion of particles carrying one negative charge ranges from 97.5% to 85.1%, and that value for particles carrying two negative charges is 0.53% - 13.6%, which is consistent with observations by Havnes et al. (Havnes, et al. 1996) and numerical results by Rapp and Lübken (Rapp and Lübken 2001). The density of positively charged particles is less than $1.1 \times 10^5$ m$^{-3}$ and is insignificant in this study.

### 3.2.2 Near the upper boundary

Next, the parameters of ice particles and plasma near the upper boundary are discussed based on the movement curves of ice particles near the upper boundary, which are shown below:

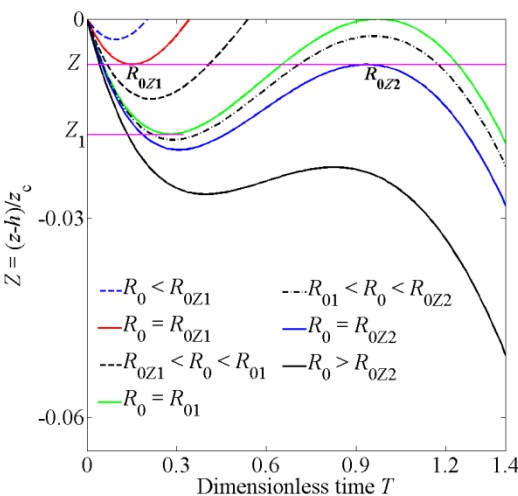

Figure 4 The movement curves of ice particles near the upper boundary. The particles with initial radius $R_{0Z1}$ move upward after turning back at the $Z$ height (the red line), and the particles with initial radius $R_{0Z2}$ move downward after turning back at $Z$ (the blue line).

For $Z_1 < Z < 0$, two kinds of particles turn back at $Z$: particles with initial radius $R_{0Z1}$ and $R_{0Z2}$. They go upward and downward separately as shown in Fig. 4. And the values of $R_{0Z1}$ and $R_{0Z2}$ are determined by equations $V_d(R_{0Z}, R_d) = 0$ and $Z(R_{0Z}, R_d) = Z$. The contribution of ice particles to the density distribution near the upper boundary can be classified as follows:

(1) $R_0 < R_{0Z1}$: ice particles cannot reach $Z$ and make no contributions to the number density.

(2) $R_{0Z1} < R_0 < R_{01}$: ice particles pass through $Z$ twice and contribute to $n_d(Z)$ twice. The radius of particles when passing through the $Z$ height can be obtained as $R_{d31}$ and $R_{d32}$ based on Eq. (27).

Meanwhile their corresponding velocities are calculated as $V_{d31}$ and $V_{d32}$ respectively based on Eq. (25).

(3) $R_{01} < R_0 < R_{0Z2}$: ice particles pass through $Z$ three times. The corresponding radii and velocities at $Z$ are defined as $R_{d41}$, $R_{d42}$, $R_{d43}$; $V_{d41}$, $V_{d42}$, $V_{d43}$.

(4) $R_0 > R_{0Z2}$: ice particles pass through $Z$ only once and their radius and velocity are $R_{d5}$ and $V_{d5}$, respectively.

Substituting these parameters into Eq. (28) and (29), the density and mean radius of ice particles in the range $Z_1 < Z < 0$ are deduced as

$$n_d(Z) = n_0 \int_{R_{0Z1}}^{R_{01}} |V_{d0}| F(R_0) \left[ \frac{1}{|V_{d31}(R_0, R_{d31})|} + \frac{1}{V_{d32}(R_0, R_{d32})} \right] dR_0$$

$$+ n_0 \int_{R_{01}}^{R_{0Z2}} |V_{d0}| F(R_0) \left[ \frac{1}{|V_{d41}(R_0, R_{d41})|} + \frac{1}{V_{d42}(R_0, R_{d42})} + \frac{1}{|V_{d43}(R_0, R_{d43})|} \right] dR_0 \quad (34)$$

$$+ n_0 \int_{R_{0Z2}}^{R_{0max}} \frac{|V_{d0}| F(R_0)}{|V_{d5}(R_0, R_{d5})|} dR_0$$

$$\bar{R}_d(Z) = \frac{n_0}{n_d(Z)} \int_{R_{0Z1}}^{R_{01}} |V_{d0}| F(R_0) \left[ \frac{R_{d31}}{|V_{d31}(R_0, R_{d31})|} + \frac{R_{d32}}{V_{d32}(R_0, R_{d32})} \right] dR_0$$

$$+ \frac{n_0}{n_d(Z)} \int_{R_{01}}^{R_{0Z2}} |V_{d0}| F(R_0) \left[ \frac{R_{d41}}{|V_{d41}(R_0, R_{d41})|} + \frac{R_{d42}}{V_{d42}(R_0, R_{d42})} + \frac{R_{d43}}{|V_{d43}(R_0, R_{d43})|} \right] dR_0 \quad (35)$$

$$+ \frac{n_0}{n_d(Z)} \int_{R_{0Z2}}^{R_{0max}} \frac{R_{d5} |V_{d0}| F(R_0)}{|V_{d5}(R_0, R_{d5})|} dR_0$$

The center of the radius distribution function $R_{00} = 1.08$, the characteristic width $\Delta = 0.01$, and the corresponding normalized coefficient A = 56.4.

The ice particle density in the region of $Z < Z_1$ is close to zero, since only particles with initial radius $R_0 \geq R_{01}$ can arrive at the region and the number of the particles in this radius range is very few based on the radius distribution function set above.

At the upper boundary, the number density of condensation cores $n_0$ is set as $5 \times 10^8$ m$^{-3}$; the maximum radius of condensation cores $R_{0max} = 1.3$. The number density and mean radius of ice particles are obtained from Eq. (34) and (35) and are shown in Fig. 5. Then the density distribution of electrons, ions, and charged ice particles is calculated further based on the charging model.

Figure 5(a) shows that there is a meter scale structure in the distribution of ice particle density, which is consistent with the assumed ice particle density structure scale in previous theoretical work (Lie‐Svendsen, et al. 2003;Rapp and Lübken 2003) and rocket observations (Rapp and Lübken 2004). The

average radius of ice particles is slightly larger than 5 nm (shown in Fig. 5(b)).

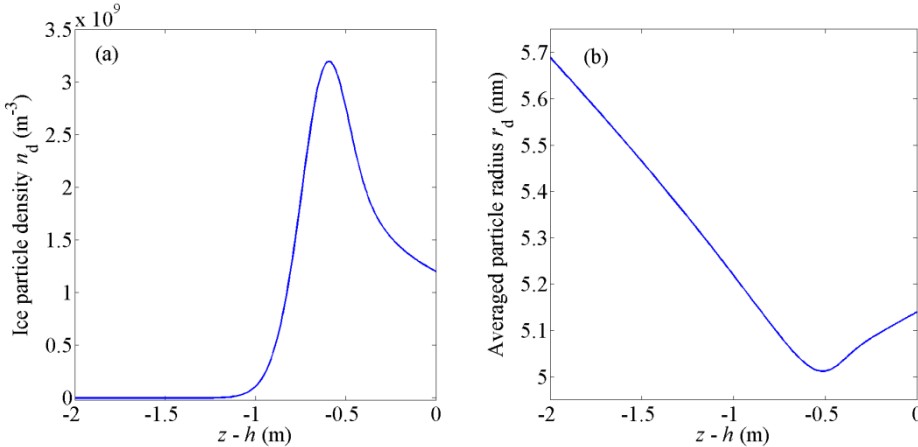

Figure 5 The distribution of (a) ice particle density and (b) the averaged particle radius near the upper boundary of condensation layer.

Figure 6(a) shows that, compared with ice particle density, there is a similar but anti-correlated structure in electron density profile because of the adsorption of electrons by particles. Due to ambipolar diffusion, ion density increases in the perturbed region. The reduction of electron density $\Delta n_e$

and the increment of ion density $\Delta n_i$ meet with the results under diffusion equilibrium approximations: $\Delta n_e \approx \Delta n_i \approx (n_{-1} + 2n_{-2})/2$, which has been concluded in reference (Lie‑Svendsen, et al. 2003). From Fig. 6(c) and Fig. 6(d), it can be seen that, 97% of the particles carry one negative charge, and particles carrying two negative charges are very few. This is reasonable for particles with radius slightly larger than 5 nanometers.

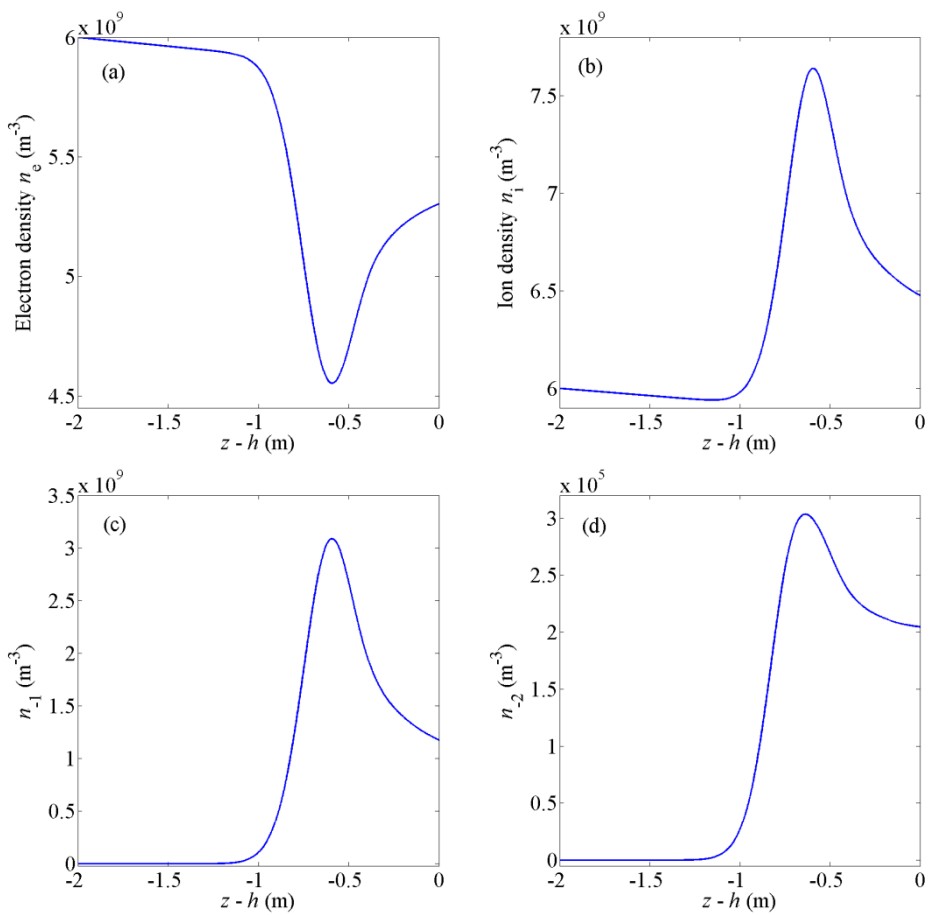

Figure 6 The number density distribution of (a) electrons, (b) ions, (c) particles carrying one negative charge, and (d) particles carrying two negative charges near the upper boundary of condensation layer at $t = 1000$ s.

## 3.3 The influence of the vertical wind speed on the spatial scale of the irregularities

We vary the vertical wind speed from 3 to 5 cm/s to investigate the influence of the wind speed on the spatial scale of the irregularities. With other parameters remaining the same, the numerical results are shown in Fig. 7 and Fig. 8. With the increase of wind speed, the spatial scale of the irregularities increases, because lager wind speed corresponds to larger critical particle radius $r_c$ (see Eq. (9)) in the growth model, which further leads to longer time scale ($t_c$) and larger spatial scale ($z_c$) of ice particle growth and movement. In addition, as shown in Fig. 7(c) and Fig. 7(d), with the increase of the wind speed, the variation amplitude of electron density and ion density near the lower boundary increases obviously. This is because the averaged radius of the ice particles increases with the extension of

particle growth time (see Fig. 7(b)), and the particles' influence on the plasma increases. And the variation amplitude of electron density and ion density near the upper boundary does not change much (see Fig. 8(c) and Fig. 8(d)), because the averaged radii of the ice particles for different wind speed don't differ very much with each other, as shown in Fig. 8(b).

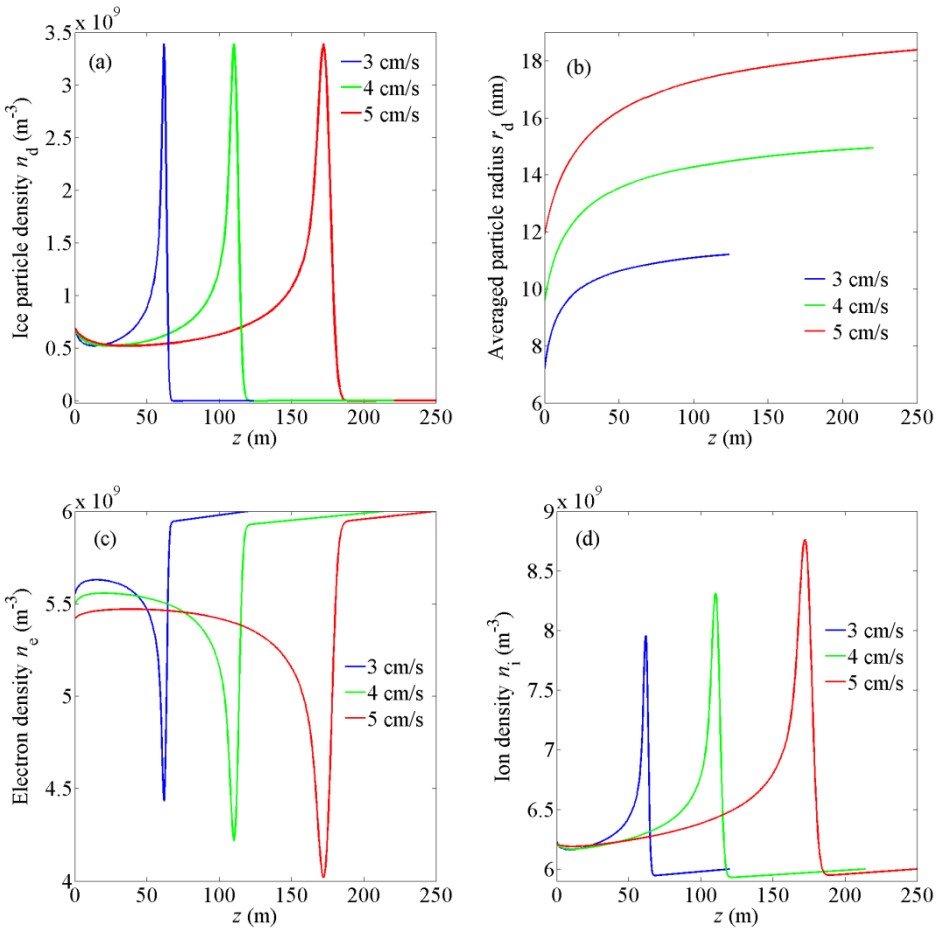

Figure 7 The distribution of (a) ice particle density, (b) the averaged particle radius, (c) electron density, and (d) ion density for various vertical wind speeds near the lower boundary of condensation layer.

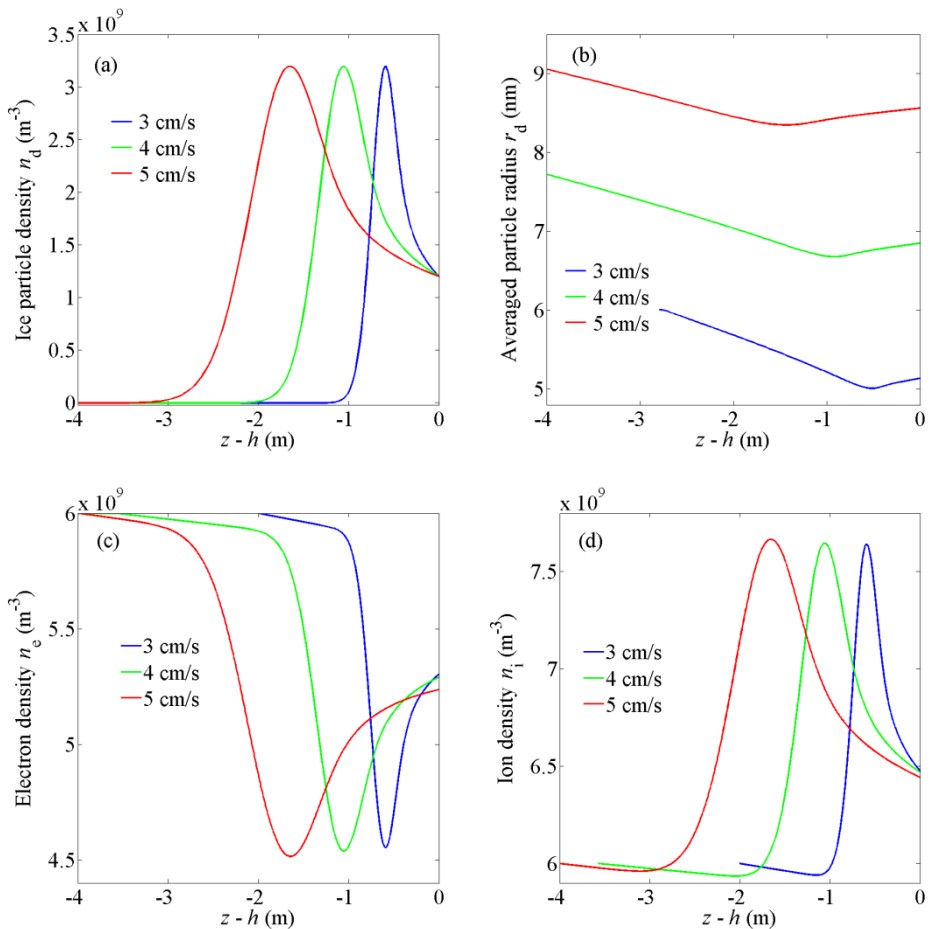

Figure 8 The distribution of (a) ice particle density, (b) the averaged particle radius, (c) electron density, and (d) ion density for various vertical wind speeds near the upper boundary of condensation layer.

### 3.4 The influence of the water vapor density on the spatial scale of the irregularities

The water vapor density can also affect the spatial scale of the particle density structures by affecting the change rate of particle radius. It is illustrated in Fig. 9 and Fig. 10 that the spatial scale of the irregularities decreases when the water vapor density increases. A lager vapor density results in a lager change rate of particle radius (see Eq. (10)) and a shorter time scale ($t_c$) of ice particle growth. Then the particles can reach the inversion condition faster and the reverse position is closer to the boundary, which means the spatial scale of the ice particle density structures gets shorter.

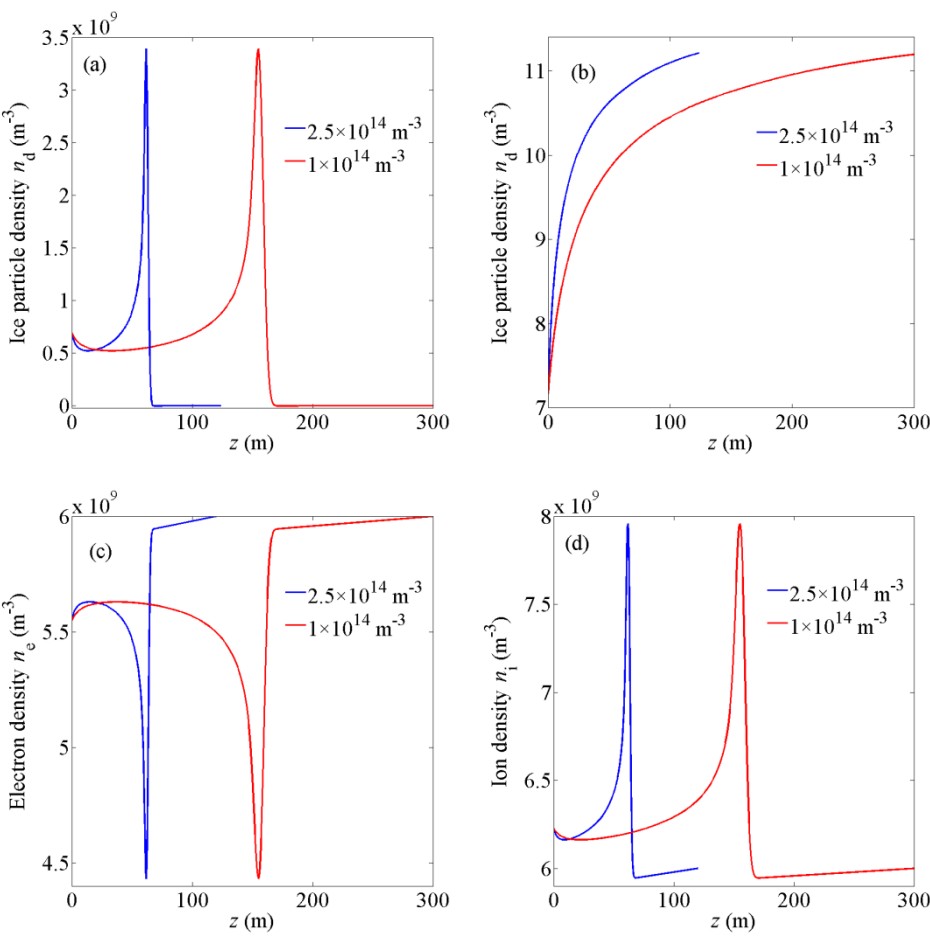

Figure 9 The distribution of (a) ice particle density, (b) the averaged particle radius, (c) electron density, and (d) ion density for various water vapor densities near the lower boundary of condensation layer.

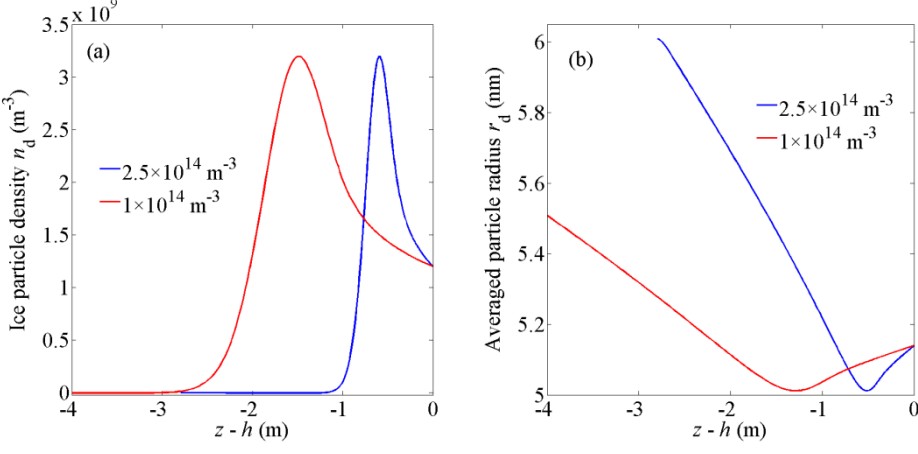

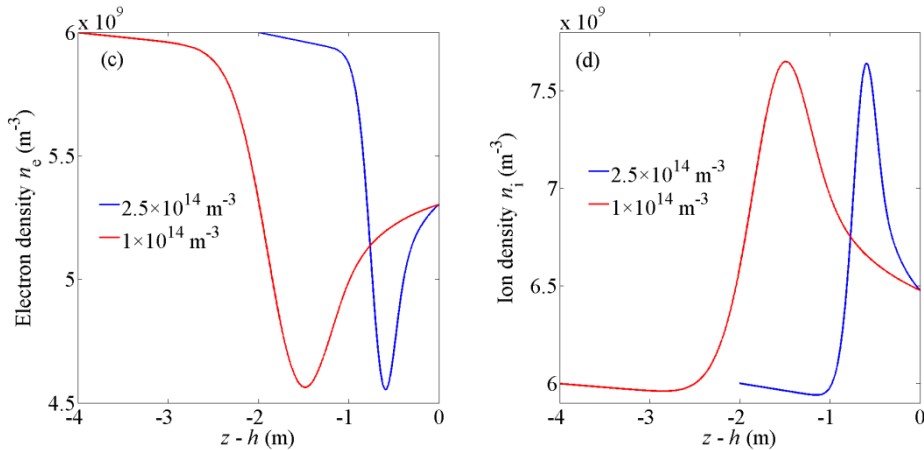

Figure 10 The distribution of (a) ice particle density, (b) the averaged particle radius, (c) electron density, and (d) ion density for various water vapor densities near the upper boundary of condensation layer.

## 3.5 The influence of the altitude on the irregularities spatial scale

In this subsection, we will discuss the effect of the altitude on the spatial scale of the irregularities. The altitude mainly affects the neutral gas density $n_n$, ion composition, ion mass $m_i$, the production rate for plasma $Q$, the electron-ion recombination coefficient $\alpha$, and the plasma density $n_0$ without ice particles. Besides 85 km, we choose 82 and 88 km altitude, which are near the lower and upper limits of the PMSE region (Lie‐Svendsen, et al. 2003). According to references (Blix 1999;Lübken 1999;Rapp and Lübken 2001), at 82 km, the positive ions are mainly $(H_3O)^+(H_2O)_3$ cluster ions with $m_i = 73\ m_u$, and other parameters are set as: $n_n = 4.2 \times 10^{20}$ m$^{-3}$, $Q = 6.3 \times 10^7$ m$^{-3}$s$^{-1}$, $\alpha = 7 \times 10^{-12}$ m$^3$s$^{-1}$ and $n_0 = 3 \times 10^9$ m$^{-3}$; at 88 km altitude, the positive ions are mainly $NO^+$ with $m_i = 30\ m_u$, and other parameters are: $n_n = 1.1 \times 10^{20}$ m$^{-3}$, $Q = 6 \times 10^7$ m$^{-3}$s$^{-1}$ $\alpha = 6 \times 10^{-13}$ m$^3$s$^{-1}$, and $n_0 = 1 \times 10^{10}$ m$^{-3}$. Fig. 11 and Fig. 12 present the numerical results. Since the ambient plasma density $n_0$ is so different at different altitudes, we have calculated the electron density relative change $\Delta n_e/n_e$ and the ion density relative change $\Delta n_i/n_i$ for comparison's sake, where $\Delta n_e = n_e - n_0$ and $\Delta n_i = n_i - n_0$. Fig. 11(a) and Fig. 12(a) show that, with the increase of altitude, the spatial scale of the ice particle density irregularities becomes shorter. The reason is that higher altitude corresponds smaller neutral density $n_n$ and smaller critical particle radius $r_c$ (see Eq. (9)), which further leads to shorter time scale ($t_c$) and spatial scale ($z_c$) of ice particle growth and movement. It is remarkable that the spatial scale of the electron density irregularities at lower altitude is

longer than that at higher altitude (see Fig. 11(c) and Fig. 12(c)), which agrees with Bremer et al.'s view on explaining the phenomenon that, at lower altitude, the PMSE signals detected by long-wavelength radar (half wavelength = 54 m) are stronger than those detected by short-wavelength radar (half wavelength = 2.8 m) (Bremer, et al. 1997). In addition, with the increase of the altitude, the relative change amplitude of electron density and ion density decreases significantly, because the averaged radius of the ice particles at higher altitude is smaller and the influence of ice particles on plasma decreases.

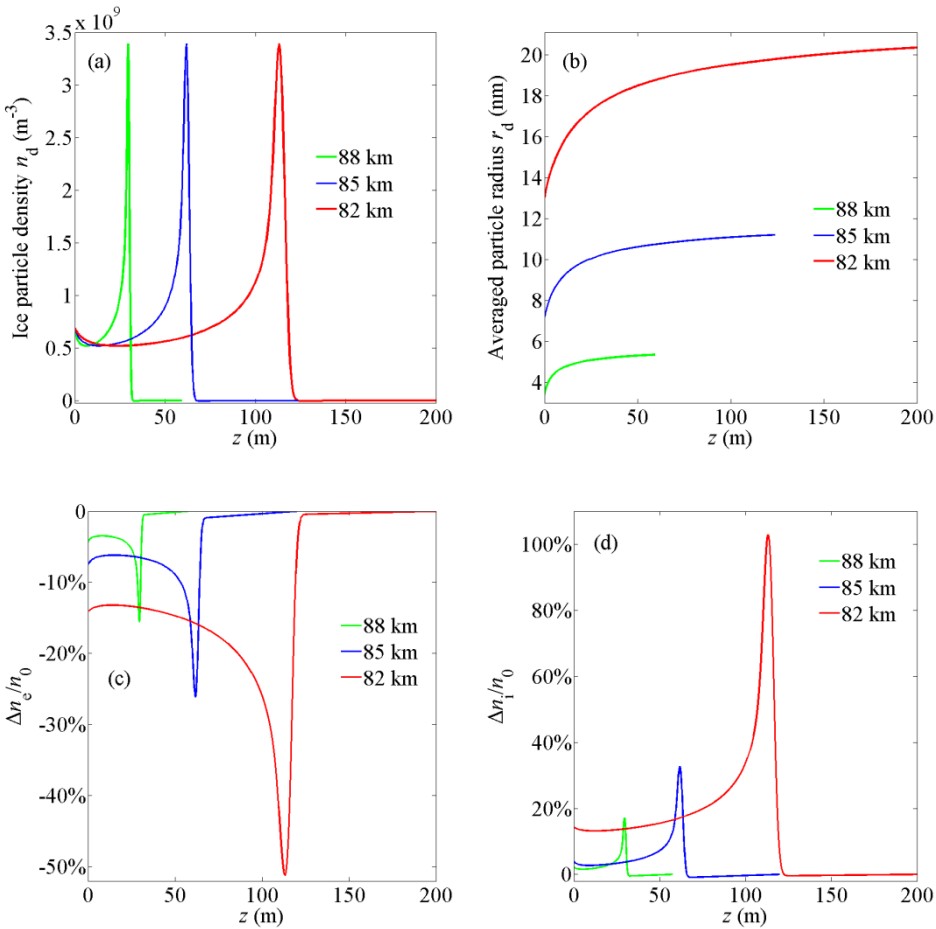

Figure 11 The distribution of (a) ice particle density, (b) the averaged particle radius, (c) the relative change of electron density $\Delta n_e/n_e$, and (d) the relative change of ion density $\Delta n_i/n_i$ at various altitudes near the lower boundary of condensation layer.

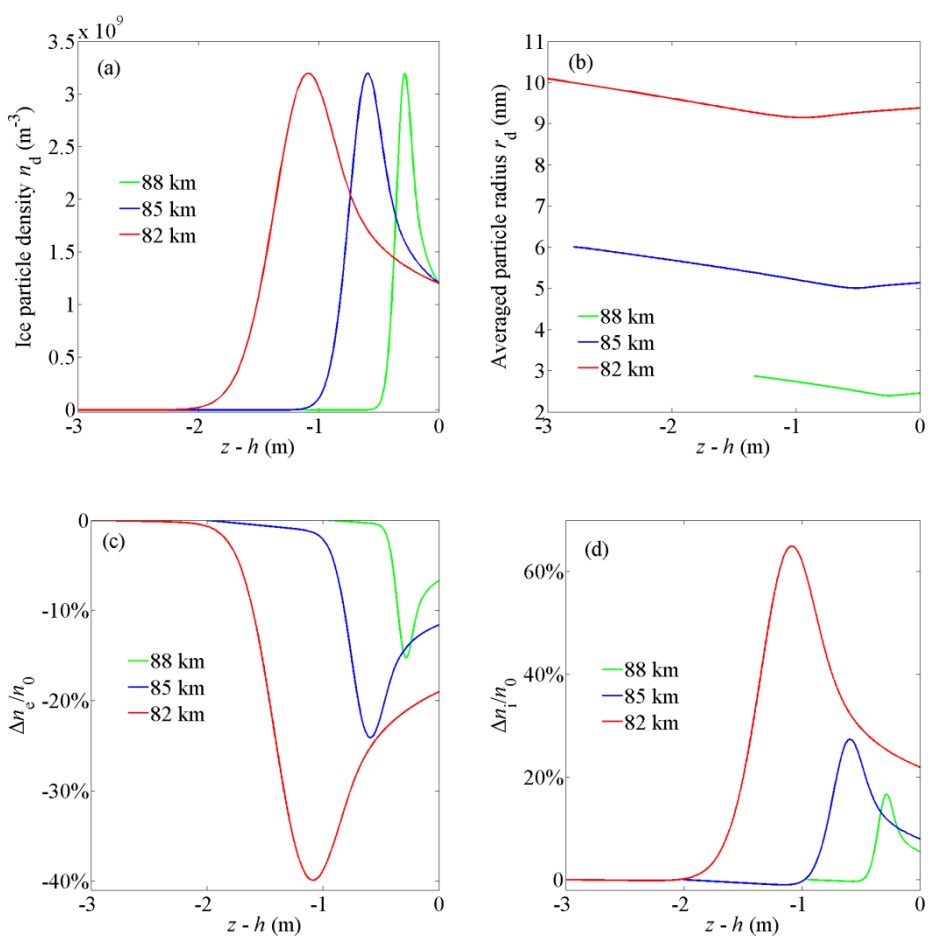

Figure 12 The distribution of (a) ice particle density, (b) the averaged particle radius, (c) the relative change of electron density $\Delta n_e/n_e$, and (d) the relative change of ion density $\Delta n_i/n_i$ at various altitudes near the upper boundary of condensation layer.

## 4 Conclusions

In summary, a growth and motion model of ice particles is originally developed based on the equation of motion of a variable mass object to explain the formation of ice particle density irregularities with meter scale in the polar mesopause region. The density profile of ice particles with height is investigated according to the conservation of particle number. Based on the growth and motion model, the small-scale structures of ice particle density are produced successfully. And then the density distributions of electrons and ions corresponding to the ice particle density distribution are obtained based on the quasi-neutrality and the discrete charging model. The more detailed conclusions are shown

as follow.

The ice particle radius increases linearly with time. But there is a complex relation between the velocity and radius of particles due to the variable mass of ice particles and complicated force on them. And for a certain radius of the condensation nucleus, ice particles can bounce near the boundary layer, which leads to the local gathering phenomenon of ice particles and meter scale ice particle density structures are produced. The spatial scale of the density structures can be affected by the vertical wind speed, water vapor density, and altitude. The spatial scale increases with the increase of wind speed, and decreases with the increase of the water vapor density and altitude. And the small-scale ice particle density irregularities can remain stable if these atmospheric conditions do not change. In the ice particle gathering region, the electron density is anti-correlated to the charged ice particle density and the ion density because of the plasma attachment by ice particles and plasma diffusion. To sum up, the small-scale ice particle density irregularities are formed and maintained in polar mesopause region based on the growth and motion model, and the corresponding small-scale electron density structures are in accordance with most rocket observations.

**Acknowledgements**

The research has been financially supported by the National Natural Science Foundation of China under Grant Nos. 11775062 and 61601419 and the Key Laboratory Foundation of National Key Laboratory of Electromagnetic Environment under Grant No. 614240319010303.

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
