# Peer review of "The research on small-scale structures of ice particle density and electron density in the mesopause region"

_Annales Geophysicae, 2019_

## Referee Comment (RC1) · Anonymous Referee #1 · 12 Apr 2019

This manuscript describes development of a model and associated calculations for ultimately determining the ice particle and electron density in the mesopause region. The electron density structures are particularly important for producing Polar Mesospheric Summer Echoes PMSEs and one ultimate goal of this work is to contribute to an understanding of the PMSE source region. The model utilizes a growth model for the ice particles (collision and adsorption of water vapor and condensation nuclei), and a velocity model (dependent on the ice particle mass and dependent on gravity and neutral drag forces) to ultimately determine the ice particle density with altitude. A charging model (OML with CEC) and quasi-neutrality is then used to determine the electron density knowing the ice particle density. Results of using this model are used to show

a reduction in electron density in the source region. These reductions produce radar scatter associated with PMSE.

The manuscript is relatively well organized and well laid out. There are some issues with English grammar and style that clearly should be addressed (there is not an unreasonably large number of these English issues, however) . However, there are some serious issues that preclude publication in Annales Geophysicae AG at this time. A key issue is that the authors have not made a persuasive case of the contribution to the field of this work. They have presented a model and some calculations but not effective tie these to observations to lend credibility to the model results. Also they have not articulated a well-defined, focused issue in the field they want to address. There has been past work in this field with previous models. There is no substantive discussion on how their model is an improvement over past models and what unresolved issues they have been able to solve that past models have not. Therefore, the paper is not suitable for publication in AG in its current form. There must be major revisions and the authors must address these key issues. Further detail of some of the critical weaknesses are as follows:

1. The last sentence (line 23-25) of the Abstract is indicative of the major problem. This sentence is vague. Why is this work important ? The rest of the abstract has not made a case for this. In fact, the last sentence is very well known to be the case from other work! No novelty of this work is stated.

2. The authors mention another well-known work in this field (Lie-Svenson et al. 2003). How is this work an advance over the past work ? This should at least be clearly shown since Lie-Svenson is often used as a benchmark work. Also, the work of Lie-Svenson shows the importance of using ion mass (through the ion continuity equation) on the electron and ion structures in the PMSE source region. The work has been validated through experimental observations. Some of these effects has been described by the work of A. Mahmoudian, On the signature of positively charged dust particles on plasma irregularities in the mesosphere, J. Atmos. Sol. Terr. Phys., 2013 which is

based on earlier work by Chen and Scales, JGR 2005. Therefore, this implies the authors work is not be consistent with observations since it does not contain ion inertia (it just assumes the Boltzmann approximation) ? No direct substantive comparison with data has been shown in this work to lend any validity.

3. What inaccuracies are introduced into the model due to the fact that an equilibrium charge is considered (equation 22). Lie-Svenson et al and other work consider a dynamically time varying particle charge. This would appear to be particularly important since the ice particle mass/radius is changing.

4. In the model section 2, there appears to be too much detail when the primary equation for the ice particle velocity model is equation 8 (perhaps equation 1 should be stated for completeness). The rest of the approximations may be useful but they can be much more succinctly summarized to shorten this section and eliminate all the equations. The final simplified collision equations may also be useful.

5. In general, one could strongly argue that the plasma (and charging) is much less well modeled in the model equations in section 2 than previous models (ie. Lie-Svenson et al., Chen and Scales). Therefore, it is highly questionable if the current work is an advance since there is no comparison using these past modeling approaches. This, again, goes back to the key issue with the manuscript.

6. The model results in Section 3 show some promising trends but these must be more closely compared to observational data. Also, there appear to be no direct linkages to a specific observation the authors are trying to understand. The authors should strive to do more than demonstrate their model does what is expected from the basic physics. Only general comparisons are made to observations which is not enough for a novel contribution.

7. Again, the authors should strive to see if their model is consistent with observations. For example, the average number of charges is less than one (see line 264) with values of 0.2 and 0.3. Does this indicate that the charging model (using a simple

equilibrium charge) is insufficient ? Doesn't the particle growth impact what charging model is used. Does the fact that the average charge is less than 1.0 indicate there are positive, negative, and uncharged particles? This has been observed/postulated during experiments? The current simple OLM equilibrium charging model does not take the fact of dynamic particle growth into consideration and may likely be inadequate for what the authors are trying to do (with such small initial particle sizes). This has not been commented on at all. For such low particle charges would a stochastic model (e.g. Mahmoudian) be better.

8. Figure 3 and 4 appear to show the electron density structures. These appear to be on the space scale of 10 meters or less. How do these results compare with other models, e.g. Lie-Svenson et al. Also why are these results an advance over these past modeling results?

Summary: This manuscript is not suitable for publication in AG at this time. If the authors consider a revision (which should be major) the key points the authors should consider are:

1. Making stronger case for why this work is superior to past models (i.e. Lie-Svenson). Certainly the author's model is inferior in terms of the model of the ionospheric plasma (no ion inertia) and charging (no dynamical variation) model. A possible advantage is the ice particle growth model but this would appear to be problematic as well without properly doing the charging model correctly. If the novelty in the ice particle growth does not counterbalance the weakness in plasma and charging models, then there is no real contribution or advance in the modeling.

2. There is no substantive comparison with observational data or a focus of an important unresolved scientific issue addressed. This was not clearly articulated and again is a substantial weakness in the paper. It should be addressed in a summary/discussion section and also noted in the Abstract.

ANGEOD

Interactive
comment

---

## Referee Comment (RC2) · Anonymous Referee #2 · 22 Jun 2019

This paper presents a model where they investigate whether gravity, the neutral drag force, and ice particle growth by adsorption of water vapor can explain why ice particles near the polar mesospause are frequently seen to be confined into small-scale structures in summer. Much has now been understood about these ice particles, and we know how, once these small-scale structures have been created, the polar mesosphere summer radar echoes (PMSE) arise. However, we still do not have a good understanding of the formation mechanism of these small-scale structures, which this paper aims to improve. I therefore think that a paper on this topic is well worth publishing. However, I do have some minor questions regarding their model, which should be resolved before this paper is considered for publication. 1. In line 126, "Substituting

[Figure]

Eq. (9) into Eq. (2)..." should be changed to "Substituting Eq. (9) into Eq. (3)..." 2. In line 133-134, there are two predicates in the sentence "It is set that z0 = 0 for the lower boundary and z0 = h for the upper one, here h is the distance between the two boundaries." 3. Authors should add some new references showing new progress in pmse.

―――――――――――――――――――

---

## Author Comment (AC1) · 26 Jun 2019

Dear Reviewer, Thank you for your insightful comments concerning our manuscript entitled "The research on small-scale structures of ice particle density and electron density in the mesopause region". Those comments are all valuable and very helpful for revising and improving our paper, as well as the important guiding significance to our researches. We have studied these comments carefully and have made corrections which we hope make our paper more acceptable. The responds to the comments are as following. Once again, special thanks to you for good comments and hope that the correction will meet with approval.

[Figure]

Responses to Reviewer

This manuscript describes development of a model and associated calculations for ultimately determining the ice particle and electron density in the mesopause region. The electron density structures are particularly important for producing Polar Mesospheric Summer Echoes PMSEs and one ultimate goal of this work is to contribute to an understanding of the PMSE source region. The model utilizes a growth model for the ice particles (collision and adsorption of water vapor and condensation nuclei), and a velocity model (dependent on the ice particle mass and dependent on gravity and neutral drag forces) to ultimately determine the ice particle density with altitude. A charging model (OML with CEC) and quasi-neutrality is then used to determine the electron density knowing the ice particle density. Results of using this model are used to show a reduction in electron density in the source region. These reductions produce radar scatter associated with PMSE.

The manuscript is relatively well organized and well laid out. There are some issues with English grammar and style that clearly should be addressed (there is not an unreasonably large number of these English issues, however).

Response

Thank you very much for pointing it out. We have gone over the text and some English usage and grammar mistakes have been revised to make it easier to understand. However, there are some serious issues that preclude publication in Annales Geophysicae AG at this time. A key issue is that the authors have not made a persuasive case of the contribution to the field of this work. They have presented a model and some calculations but not effective tie these to observations to lend credibility to the model results. Also they have not articulated a well-defined, focused issue in the field they want to address. There has been past work in this field with previous models. There is no substantive discussion on how their model is an improvement over past models and what unresolved issues they have been able to solve that past models have not.

Response

Thank you very much for your valuable and thoughtful comments. It is believed that small scale electron density fluctuations can cause PMSE phenomenon (Rapp and Lübken 2004). And previous works (Lie‐Svendsen, et al. 2003;Rapp and Lübken 2003) have shown that ice particle irregularities on meter scale can create electron density fluctuations on the similar scale due to plasma attachment by particles and plasma diffusion. In their models, the ice particle density profile is given directly, with an embedded small scale Gaussian structure. However, the formation mechanism of these small-scale particle density structures has not been fully understood. In view of this, the aim of our study is trying to explain the formation of these ice particle irregularities through the growth and movement model. The analysis of relevant previous work and the purpose of this paper have been added in the introduction. Meanwhile, to make our model results more accurate and credible, we have modified the plasma model according to the detailed comments below, which includes dynamic continuity equations for ice particles with various charges and ions, momentum equation for ions and electrons, and quasi-neutral condition. The results of the revised model are in agreement with previous work by Lie-Svenson et al. (Lie‐Svendsen, et al. 2003), e.g., for particles with radii of 11 nm or less, electron density is anti-correlated to charged ice particle density and ion density, which is in line with most rocket observations. We have modified the charging model in the second section, and have added a comparison with previous work in the third section.

Therefore, the paper is not suitable for publication in AG in its current form. There must be major revisions and the authors must address these key issues. Further details of some of the critical weaknesses are as follows:

1. The last sentence (line 23-25) of the Abstract is indicative of the major problem. This sentence is vague. Why is this work important? The rest of the abstract has not made a case for this. In fact, the last sentence is very well known to be the case from other work! No novelty of this work is stated.

Response

Thank you very much for pointing it out. We are sorry for our unclearly description on the innovation and significance of this manuscript.

The main value of this paper is to propose a possible mechanism for the formation of small scale ice particle density irregularities in PMSE region based on particle growth and movement model, while the structure of ice particle density is always assumed to be some specific profiles in previous work(Chen and Scales 2005;Lie‐Svendsen, et al. 2003;Mahmoudian and Scales 2013;Rapp and Lübken 2003;Scales and Ganguli 2004). A statement of the purpose and significance of this article has been added to the abstract section.

2. The authors mention another well-known work in this field (Lie-Svenson et al. 2003). How is this work an advance over the past work? This should at least be clearly shown since Lie-Svenson is often used as a benchmark work. Also, the work of Lie-Svenson shows the importance of using ion mass (through the ion continuity equation) on the electron and ion structures in the PMSE source region. The work has been validated through experimental observations. Some of these effects have been described by the work of A. Mahmoudian, On the signature of positively charged dust particles on plasma irregularities in the mesosphere, J. Atmos. Sol. Terr. Phys., 2013 which is based on earlier work by Chen and Scales, JGR 2005. Therefore, this implies the authors work is not consistent with observations since it does not contain ion inertia (it just assumes the Boltzmann approximation)? No direct substantive comparison with data has been shown in this work to lend any validity.

Response

Thank you very much for your instructive suggestions.

Lie-Svenson et al. studied the plasma response to initially given small-scale ice particle perturbations in the mesopause region. The formation process of these small-scale

structures of ice particle density is still not fully understood. The aim of our study is trying to explain the formation of these small-scale ice particle density structures based on the growth and movement model of particles. The analysis of relevant previous work and the purpose of this paper have been added in the introduction section. After studying the previous work and observations carefully, we find that the assumption of ion immobility in our previous manuscript version was not accurate. So we modify the plasma model in the revised manuscript according to Lie-Svenson et al.'s theory (Lie‐Svendsen, et al. 2003). The revised plasma model considers production, loss and transport of ions and electrons, and dynamic particle charging. Some more detailed description on the modified plasma model has been made in the model section 2. According to the revised model, for particles with radii of 11 nm or less, electron density is anti-correlated to ion and charged ice particle density near the boundary of condensation region. It is in agreement with previous work by Lie-Svenson et al. (Lie‐Svendsen, et al. 2003) and most rocket observations(Rapp and Lübken 2004). Detailed results analysis and comparison with previous work have been added in the results and discussion section 3.

3. What inaccuracies are introduced into the model due to the fact that an equilibrium charge is considered (equation 22). Lie-Svenson et al and other work consider a dynamically time varying particle charge. This would appear to be particularly important since the ice particle mass/radius is changing.

Response

Thank you very much for pointing it out.

According to research of Lie‐Svendsen et al. (Lie‐Svendsen, et al. 2003), the assumption of chemical equilibrium would overestimate the electron depletion and seriously underestimate the ion enhancement, i.e., the equilibrium charge is indeed not a valid approximation in studying plasma response to small-scale ice particle irregularities. In view of this, we have modified the plasma model with dynamic particle charging

considered.

In our study, it is assumed that condensation nuclei enter the condensation region with a fixed flux. They grow by absorbing water vapor and move under the action of gravity and neutral drag force. Note that, the charge to mass ratio of ice particles is very low, the electric field force on the particles can be ignored compared to the other two forces, so the dynamic particle charge does not affect the formation of the final particle density profile. Ice particle density will form stable small-scale structures after several hours. The particles keep entering and leaving the condensation region, but as long as the external environment does not change, the distribution of particle density and radius remains unchanged. Then the influence of these stable small-scale structures on electron and ion density is studied by the modified charging model just like Lie‐Svendsen et al. did in their work (Lie‐Svendsen, et al. 2003).

The more detailed description of the modified plasma model has been added in the model section 2.

4. In the model section 2, there appears to be too much detail when the primary equation for the ice particle velocity model is equation 8 (perhaps equation 1 should be stated for completeness). The rest of the approximations may be useful but they can be much more succinctly summarized to shorten this section and eliminate all the equations. The final simplified collision equations may also be useful.

Response

Thank you very much for your instructive suggestions. We have summarized the approximate conditions into words to make the article more concise.

5. In general, one could strongly argue that the plasma (and charging) is much less well modeled in the model equations in section 2 than previous models (ie. Lie-Svenson et al., Chen and Scales). Therefore, it is highly questionable if the current work is an advance since there is no comparison using these past modeling approaches. This,

again, goes back to the key issue with the manuscript.

Response

Thank you very much for pointing it out. We are sorry for using a very rough plasma model in our original text. The plasma model has been modified, which considers production, loss and transport of ions and electrons, and dynamic particle charging. We have made some more detailed description on the modified plasma model in the model section 2.

The improvement of this study over previous work is to present a possible formation mechanism of small-scale ice particle structures. The analysis of relevant previous work and the research purpose of this paper have been added in the introduction section.

6. The model results in Section 3 show some promising trends but these must be more closely compared to observational data. Also, there appear to be no direct linkages to a specific observation the authors are trying to understand. The authors should strive to do more than demonstrate their model does what is expected from the basic physics. Only general comparisons are made to observations, which is not enough for a novel contribution.

Response

Thank you very much for your instructive suggestions.

The main purpose of this paper is to present a possible explanation on the formation of the small-scale ice particle irregularities in PMSE region. Through the growth model, we obtain ice particle density structure at meter scale near the boundary of condensation region, which is consistent with the assumed ice particle density structure scale in the theoretical calculations of previous work (Lie–Svendsen, et al. 2003;Rapp and Lübken 2003), and is consistent with observations by the sounding rocket flight ECT02 in July 1994 (Rapp and Lübken 2004). Based on the modified plasma model,

for particles with radii of 11 nm or less, electron density is anti-correlated to density of ions and charged ice particles, which are in agreement with rocket observations by the sounding rocket flight SCT-06 in August 1993 (Lie‐Svendsen, et al. 2003) and the sounding rocket flight ECT02 in July 1994 (Rapp and Lübken 2004), respectively.

Detailed results analysis and comparison with previous work have been added in the results and discussion section 3.

7. Again, the authors should strive to see if their model is consistent with observations. For example, the average number of charges is less than one (see line 264) with values of 0.2 and 0.3. Does this indicate that the charging model (using a simple equilibrium charge) is insufficient? Doesn't the particle growth impact what charging model is used. Does the fact that the average charge is less than 1.0 indicate there are positive, negative, and uncharged particles? This has been observed/postulated during experiments? The current simple OLM equilibrium charging model does not take the fact of dynamic particle growth into consideration and may likely be inadequate for what the authors are trying to do (with such small initial particle sizes). This has not been commented on at all. For such low particle charges would a stochastic model (e.g. Mahmoudian) be better.

Response

Thank you very much for your valuable and instructive comments.

The particle radius in this study is less than 11 nm, and an ice particle carries two negative elementary charges at most. The quantized stochastic charging model (Robertson and Sternovsky 2008) is more appropriate to determine the particle charge. Therefore, we modify the plasma model and use the quantized stochastic charging model to calculate the capture rates of electrons and ions by ice particles. The results show that for particles with a radius about 5 nm, the proportion of particles carrying one negative charge is about 97%. For particles with radii ranging from 7 nm to 11 nm, the proportion of particles carrying one negative charge ranges from 97.5% to 85.1%, and that

value for particles carrying two negative charges is in 0.53% - 13.6%, which is consistent with observations by Havnes et al. (Havnes, et al. 1996) and numerical results by Rapp and Lübken (Rapp and Lübken 2001).

As we have said before in the response to comment 3, the dynamic particle charging process does not affect the formation of the final particle density profile, i.e., the particle charging process is negligible when calculating the particle density structure based on the particle growth and motion model. After the stable particle density profile is obtained, the corresponding electron and ion density are calculated according to the modified charging model. In this case, ice particle density structure and radius keep stable, which means that the influence of dust growth and motion on charging process is negligible. More detailed results analysis and comparison with previous work have been added in the results and discussion section 3 and detailed description on the modified plasma model have been made in the model section 2.

8. Figure 3 and 4 appear to show the electron density structures. These appear to be on the space scale of 10 meters or less. How do these results compare with other models, e.g. Lie-Svenson et al. Also why are these results an advance over these past modeling results?

Response

Thank you very much for pointing it out.

The small-scale electron density structures are the consequences of ice particle density irregularities. The main improvement of this paper is to propose a possible formation mechanism of the ice particle density irregularities based on particle growth and movement model, while previous work directly sets the particle density structure to a specific form. The scale and position of the ice particle density irregularities are affected by particle radius distribution function, neutral wind speed, and water vapor density etc. For example, the particle density profiles for different radius distribution functions are shown in Fig. 1.

Caption of figure 1: Figure 1 The ice particle density distribution near the (a) upper boundary and (b) lower boundary of the condensation layer for different radius distribution functions. In (a) the center of the radius distribution function R00 = 1.08. In (b) R00 = 0.8. The solid blue line: $\Delta$ = 0.01 and A = 56.4; the red dotted line: $\Delta$ = 0.03 and A = 18.8.

Summary: This manuscript is not suitable for publication in AG at this time. If the authors consider a revision (which should be major) the key points the authors should consider are:

1. Making stronger case for why this work is superior to past models (i.e. Lie-Svenson). Certainly the author's model is inferior in terms of the model of the ionospheric plasma (no ion inertia) and charging (no dynamical variation) model. A possible advantage is the ice particle growth model but this would appear to be problematic as well without properly doing the charging model correctly. If the novelty in the ice particle growth does not counterbalance the weakness in plasma and charging models, then there is no real contribution or advance in the modeling.

Response

Thank you very much for your instructive suggestions.

The main improvement of this paper is to propose a possible mechanism for the formation of small-scale ice particle density irregularities based on particle growth and movement model, while the particle density structure in previous work was always assumed as some specific forms.

After consulting previous work and observations, we find that the assumption of ion immobility in our original manuscript is not accurate and the equilibrium charge is not a valid approximation for studying plasma response to small-scale ice particle irregularities. So we modified the plasma model used in this paper by considering the production, loss and transport of ions and electrons, and dynamic particle charging

processes.

2. There is no substantive comparison with observational data or a focus of an important unresolved scientific issue addressed. This was not clearly articulated and again is a substantial weakness in the paper. It should be addressed in a summary/discussion section and also noted in the Abstract.

Response

Thank you very much for pointing it out. We are sorry for not comparing the results with the observations.

The modified model shows that, for particles with radii of 11 nm or less, the electron density is anti-correlated to ion and charged ice particle density, which is in line with rocket observations by the sounding rocket flight SCT-06 in August 1993 (Lie‐Svendsen, et al. 2003) and the sounding rocket flight ECT02 in July 1994 (Rapp and Lübken 2004), respectively. We have added more detailed results analysis and comparison with previous work in the results and discussion section 3.

The main purpose of this paper is to present a possible explanation of the origin of the small-scale ice particle irregularities in PMSE region. Previous works (Lie‐Svendsen, et al. 2003;Rapp and Lübken 2003) have shown that ice particle density irregularities on meter scale can create electron density fluctuations on the similar scale, which can cause PMSE phenomenon. In their models, however, the ice particle density profile is given initially, such as small scale Gaussian structure. The aim of our study is trying to present a possible explanation on the formation of these ice particle irregularities through the growth and movement model. The analysis of relevant previous work and the research purpose of this paper have been added in the introduction to make the paper more coherent. Also, a statement of the purpose and value of this article has been added to the abstract section.

REFERENCE

Chen C., Scales W.: Electron temperature enhancement effects on plasma irregularities associated with charged dust in the Earth's mesosphere, Journal of Geophysical Research: Space Physics, 110, 2005.

Havnes O., Trøim J., Blix T., etc.: First detection of charged dust particles in the Earth's mesosphere, Journal of Geophysical Research: Space Physics, 101, 10839-10847, 1996.

Lie‐Svendsen Ø., Blix T., Hoppe U. P., etc.: Modeling the plasma response to small‐scale aerosol particle perturbations in the mesopause region, Journal of Geophysical Research: Atmospheres, 108, 8442, 2003.

Mahmoudian A., Scales W.: On the signature of positively charged dust particles on plasma irregularities in the mesosphere, Journal of Atmospheric and Solar-Terrestrial Physics, 104, 260-269, 2013.

Rapp M., Lübken F.-J.: Modelling of particle charging in the polar summer mesosphere: Part 1—General results, Journal of Atmospheric and Solar-Terrestrial Physics, 63, 759-770, 2001. Rapp M., Lübken F.-J.: Polar mesosphere summer echoes (PMSE): Review of observations and current understanding, Atmospheric Chemistry and Physics, 4, 2601-2633, 2004.

Rapp M., Lübken F. J.: On the nature of PMSE: Electron diffusion in the vicinity of charged particles revisited, Journal of Geophysical Research: Atmospheres, 108, 8437, 2003.

Robertson S., Sternovsky Z.: Effect of the induced-dipole force on charging rates of aerosol particles, Physics of Plasmas, 15, 040702, 2008.

Scales W., Ganguli G.: Investigation of plasma irregularity sources associated with charged dust in the Earth's mesosphere, Advances in Space Research, 34, 2402-2408, 2004.
Please also note the supplement to this comment:
https://www.ann-geophys-discuss.net/angeo-2019-10/angeo-2019-10-AC1-supplement.pdf
* * *
[Figure]

**Fig. 1.** Figure 1 The ice particle density distribution near the (a) upper boundary and (b) lower boundary of the condensation layer for different radius distribution functions.

---

## Author Comment (AC2) · 26 Jun 2019

Dear Reviewer, Thank you for your insightful comments concerning our manuscript entitled "The research on small-scale structures of ice particle density and electron density in the mesopause region". Those comments are all valuable and very helpful for revising and improving our paper, as well as the important guiding significance to our researches. We have studied these comments carefully and have made corrections which we hope make our paper more acceptable. The responds to the comments are as following. Once again, special thanks to you for good comments and hope that the correction will meet with approval.

[Figure]

Responses to Reviewer

This paper presents a model where they investigate whether gravity, the neutral drag force, and ice particle growth by adsorption of water vapor can explain why ice particles near the polar mesospause are frequently seen to be confined into small-scale structures in summer. Much has now been understood about these ice particles, and we know how, once these small-scale structures have been created, the polar mesosphere summer radar echoes (PMSE) arise. However, we still do not have a good understanding of the formation mechanism of these small-scale structures, which this paper aims to improve. I therefore think that a paper on this topic is well worth publishing. However, I do have some minor questions regarding their model, which should be resolved before this paper is considered for publication. 1. In line 126, "Substituting Eq. (9) into Eq. (2)..." should be changed to "Substituting Eq. (9) into Eq. (3)..."

Response

Thank you very much for pointing it out. We have corrected this incorrect description.

2. In line 133-134, there are two predicates in the sentence "It is set that $z0 = 0$ for the lower boundary and $z0 = h$ for the upper one, here h is the distance between the two boundaries."

Response

Thank you very much for pointing it out. We have corrected this grammatical error and gone over the text further. Some English usage and grammar mistakes have been revised to make it easier to understand.

3. Authors should add some new references showing new progress in pmse.

Response

Thank you very much for your instructive suggestions. We have added more references about influence of small-scale particle density structures on plasma in PMSE region in

the introduction.
* * *

---

## Author Response (AR2)

Dear Dr. Andrew J. Kavanagh and Dr. Christoph Jacobi,

On behalf of my co-authors, we thank you very much for giving us another opportunity to revise our manuscript entitled "The research on small-scale structures of ice particle density and electron density in the mesopause region" (#angeo-2019-10). We would like to express our great appreciation to you and the reviewer for some constructive comments and suggestions on our manuscript. Based on the comments and requests, we have made careful modification on the original manuscript. We attached revised manuscript and every question from the reviewer was summarized.

Looking forward to hearing from you.

Thank you and best regards!

Yours sincerely

Ruihuan Tian

Dear Reviewer,

Thank you for your insightful comments concerning our manuscript entitled "The research on small-scale structures of ice particle density and electron density in the mesopause region". Those comments are all valuable and very helpful for revising and improving our paper, as well as the important guiding significance to our researches. We have studied these comments carefully and have made corrections which we hope make our paper more acceptable. The responds to the comments are as following. Once again, special thanks to you for good comments and hope that the correction will meet with approval.

**Responses to Reviewer**

The authors have made significant revisions to the original manuscript (particularly the ionospheric plasma model) that have strengthened it significantly. They are to be credited/acknowledged for this. However, the prime issue with the first revision still exists. The authors have not made a persuasive case of the advance made by this work relative to other work in the field. This is not clarified in the Abstract, Introduction, or most importantly, the Conclusions section. Therefore, the manuscript is still not worthy of publication in its current state until this critical issue is addressed through further major revisions.

Response

Thank you very much for pointing it out. We are sorry for not giving a persuasive description on the novelty of our work.

To sum up, the theoretical model of this manuscript mainly consists of two parts: one is the particle growth and motion model that determines the ice particle density irregularities, and another is the plasma model that determines the corresponding electron density irregularities.

Between the two theoretical models, the particle growth and motion model is originally developed by our research group to try to give a possible explanation on the formation of the ice particle density irregularities in PMSE region. The motion

equation of variable mass object is introduced into the particle growth and motion model, and as far as we know, similar work hasn't been reported before. Although the particle growth and motion model may be not perfect, we still find out some useful results and the significance of this work shouldn't be denied. As for the plasma model for determining the corresponding electron density irregularities, it has been studied by many authors, and we utilize the well-developed and widely used one in reference (Lie‑Svendsen et al. 2003) to conduct the calculation of electron density irregularities corresponding to the obtained particle irregularities to make our paper more complete.

Frankly speaking, the particle growth and motion model is the emphasis of this manuscript and the place where the novelty lies. We have made some modification in the manuscript to highlight the advances of our work.

Some examples of this inadequacy are as follows:

1. Abstract: The authors do not note the uniqueness or novelty of the growth model. Is it the first of its kind or an advance over previous models? This must be clarified to make the case that this is work suitable for publication in Annales Geophysicae. The last few sentences state the results of applying the plasma model to the neutral density obtained with the model. The conclusion is correct, but is it also well known by a number of previous investigations (both modeling and data). Again, the question arises what is the novelty or contribution of the work. The reader is again brought back to the fact that the neutral model (and incorporating it into the plasma model) is most likely the primary contribution of the manuscript. However, this is poorly articulated and a persuasive case not made.

Response

Thank you very much for pointing it out. We are sorry for our unclearly description on the novelty of this manuscript.

In this manuscript, we have proven that small-scale density structures of ice particle can be obtained in PMSE region via the particle growth and motion model. This

model is originally developed by our research group by combining action of gravity, neutral drag force and particle growth by adsorption of water vapor. We believe that our work should have some novelty since the small-scale structures of ice particle density can be produced by our model.

And calculating the influence of these obtained ice particle density structures on plasma is actually not the focus of this paper, but makes it more complete, because the small-scale electron density irregularities (but not the ice particle density irregularities) are the direct cause of PMSE.

We have made some modification in Abstract to highlight the novelty of this paper.

2. Introduction: The authors state near the end of the Introduction that 'this study is trying to explain the formation of these ice particle irregularities through a growth and motion model.' Is this model new? If so, how is it different from past work? How does the new model allow the authors to solve an unknown critical problem in the field? Why is this problem important in the grand scheme of advancing the state-of-the-art in the field. Also how does incorporating this (possible new) neutral model into the ionospheric plasma model (which is the same as past models) facilitate resolution of important unresolved issues in the field. Without such a clarification, no persuasive case has been made on how the work is novel.

Response

Thank you very much for your instructive suggestions.

As far as we know, the growth and motion model of ice particles is a new model in the PMSE field since we haven't seen any similar work before.

In the polar mesopause region, there is neutral airflow moving upward. The ice particles are subjected to upward neutral drag force and downward gravity, and grow simultaneously by absorbing water vapor. In addition, the size of initial condensation nuclei has a certain distribution. These factors can cause complex trajectories of ice particles and result in an inhomogeneous distribution of particle number density, which then leads to small-scale structures of ice particle density and the

corresponding electron density structures. This may be an important mechanism that can produce PMSE phenomenon. But as far as we know, few people have studied the formation process of small-scale ice particle structures from the perspective of ice particle growth and movement. In view of this, we develop a growth and motion model of condensation nuclei in PMSE region to analyze the ice particle trajectories and calculate the number density distribution of ice particles.

In last few decades, the ice particle growth processes have been studied extensively. And most studies among of the past work are mainly concentrated on the nucleation process by considering the phase transition process (Keesee 1989; Gumbel and Witt 2002; Gumbel and Megner 2009). However, in our growth and motion model of ice particles, we assume that the supersaturated water vapor and condensation nuclei larger than the critical size already exist and stable growth of ice particles will continue when water molecules collide with them during thermal motion. We take the growth of particles into account in the equation of motion and study the complex trajectories of ice particles. Though the small-scale structures of ice particle density are produced successfully through the particle growth and motion model, we still don't dare to say that we have solved the unknown critical problem in this field since the formation mechanism of small-scale particle density structures is very complex and our model may be just one of the many mechanisms.

In addition, the neutral model and the ionospheric plasma model are separate with each other. The plasma model is carried out after the calculation of ice particle density structure. It should be stressed that the plasma model is not the emphasis of this manuscript. And we calculate the electron density irregularities based on the obtained ice particle density structures to make the manuscript more complete. So the well-developed and widely used plasma model in reference (Lie‐Svendsen et al. 2003) is employed in this manuscript to make our calculations more accurate, which should not affect the novelty of this manuscript obviously. According to the obtained plasma density structures, we can also check the validity of the growth and motion model of ice particles.

We have made some more detailed description in Introduction section to make our

models more persuasive.

3. Conclusions: The Conclusion section also does not make a persuasive case of the novelty of the work. The first paragraph, for the most part, just reiterates the model. The second paragraph should make the case for novelty of the overall contribution. It is stated 'When the radius distribution function of condensation nucleus is Gaussian, stable small-scale ice particle density structures can be obtained based on the growth and motion model.' Is this the primary contribution? If so, why is it important? Has this already been known before or have other models been able to produce this result? It is also stated 'Furthermore, the reduction of electron density and the increment of ion density are about half the charge number density of ice particles, which is in line with the results under diffusion equilibrium approximations.' Why is this important? Has it been shown with other models before? To reiterate, it is very difficult to understand the contribution or importance of the work from the Conclusion section.

Response

Thank you very much for pointing it out.

The novelty of the work is that we have originally developed the particle growth and motion model to give a possible explanation on the formation of the ice particle density irregularities in PMSE region. And it has been stressed in the new version of the manuscript.

In our particle growth and motion model, the action of gravity, neutral drag force and particle growth by adsorption of water vapor are considered. The size distribution of initial condensation nuclei can affect the width and amplitude of the final obtained ice particle density structures. No matter what kind of particle size distribution is chosen, there will be a corresponding ice particle density structures. And in this manuscript, the Gaussian distribution is used because it have been proven by U. Berger and U. von Zahn's study (Berger and Von Zahn 2002) that the size distribution of ice particles in the summer mesopause region is closer to Gaussian in shape.

As for the calculations of electron and ion densities under a certain ice particle

density structure, they are important for explaining the PMSE phenomenon and have been studied by many researchers(Lie‑Svendsen et al. 2003). But in this manuscript, they are not the novelty part and they are mainly conducted to show the electron and ion densities corresponding to our obtained ice particle structures. With the calculated electron and ion densities, we can compare them with observation results, and then the ice particle structures obtained in our manuscript can be supported by the comparison results.

The decrease of electron density $\Delta n_e$ and the increase of ion density $\Delta n_i$ in the ice particle region are similar to the results under diffusion equilibrium approximation, which has been discussed in detail in reference (Lie‑Svendsen et al. 2003). This is not an important conclusion of this manuscript, so it has been deleted for brevity.

In Conclusion section, we have made some modification to highlight the contribution of our work and make our research more persuasive.

4. Figure 3 and 6: Do these two Figures perhaps show the primary overall contribution of the work? Is the contribution the utilization of the overall model (neutral and ionospheric plasma when used together) that happens to produce results of electron fluctuations (both spatial scales and amplitudes) in line with experiments? If this is the case this needs to be carefully clarified to help address the issues raised in 1., 2., 3. Also, has the parameter regime used in Figure 3 and 6 been explored to perhaps provide a more novel contribution on unresolved questions?

**Response**

Thank you for pointing it out. We are sorry for making readers misunderstand the primary contribution of this manuscript.

As we have stated in question 2, the neutral model and the ionospheric plasma model are separate with each other. And the plasma model is carried out after the calculation of ice particle density structure. Since we have originally developed the particle growth and motion model with action of gravity, neutral drag force and particle growth by adsorption of water vapor considered, Figure 2 and 5 obtained

from the particle growth and motion model are the most important contributions of the manuscript.

Calculating the influence of these obtained ice particle density structures on plasma is not the focus of this paper, but makes it more complete, because the small-scale electron density irregularities (but not the ice particle density irregularities) are the direct cause of PMSE. Also, the obtained plasma density structures are in line with observation, which in turn supports ice particle density structures obtained in our manuscript.

In the new version of the manuscript, we have made some modifications to highlight the contribution of our work.

In overall summary, the authors again have not made the persuasive case (although it may exist) of the novelty of the work to the degree the manuscript warrants publication. They should be commended on greatly improving the ionospheric plasma model, however, this is now just equivalent to past work.

**Response**

Thank you very much for your instructive suggestions.

According to the reviewer's comments on our manuscript, we have made many modifications to highlight the novelty of our work, which is mainly focused on the particle growth and motion model. The detailed explanation on the novelty of the particle growth and motion model has been stated in question 2. Also we have made explanations for the choice of ionospheric plasma model. Since the ionospheric plasma model is not the emphasis of this manuscript, the use of the well-developed and widely used plasma model in reference (Lie‑Svendsen et al. 2003) should not affect the novelty and significance of this manuscript obviously.

[revised manuscript text omitted]
{d0}} F(R_0) \left[ \frac{1}{|V_{\mathrm{d31}}(R_0, R_{\mathrm{d31}})|} + \frac{1}{V_{\mathrm{d32}}(R_0, R_{\mathrm{d32}})} \right] \mathrm{d}R_0$$

$$+ n_0 \int_{R_{01}}^{R_{0Z2}} V_{\mathrm{d0}} F(R_0) \left[ \frac{1}{|V_{\mathrm{d41}}(R_0, R_{\mathrm{d41}})|} + \frac{1}{V_{\mathrm{d42}}(R_0, R_{\mathrm{d42}})} + \frac{1}{|V_{\mathrm{d43}}(R_0, R_{\mathrm{d43}})|} \right] \mathrm{d}R_0 \quad (34)$$

$$+ n_0 \int_{R_{0Z2}}^{R_{0\max}} \frac{V_{\mathrm{d0}} F(R_0)}{|V_{\mathrm{d5}}(R_0, R_{\mathrm{d5}})|} \mathrm{d}R_0$$

$$\bar{R}_{\mathrm{d}}(Z) = \frac{n_0}{n_{\mathrm{d}}(Z)} \int_{R_{0Z1}}^{R_{01}} V_{\mathrm{d0}} F(R_0) \left[ \frac{R_{\mathrm{d31}}}{|V_{\mathrm{d31}}(R_0, R_{\mathrm{d31}})|} + \frac{R_{\mathrm{d32}}}{V_{\mathrm{d32}}(R_0, R_{\mathrm{d32}})} \right] \mathrm{d}R_0$$

$$+ \frac{n_0}{n_{\mathrm{d}}(Z)} \int_{R_{01}}^{R_{0Z2}} V_{\mathrm{d0}} F(R_0) \left[ \frac{R_{\mathrm{d41}}}{|V_{\mathrm{d41}}(R_0, R_{\mathrm{d41}})|} + \frac{R_{\mathrm{d42}}}{V_{\mathrm{d42}}(R_0, R_{\mathrm{d42}})} + \frac{R_{\mathrm{d43}}}{|V_{\mathrm{d43}}(R_0, R_{\mathrm{d43}})|} \right] \mathrm{d}R_0 \quad (35)$$

$$+ \frac{n_0}{n_{\mathrm{d}}(Z)} \int_{R_{0Z2}}^{R_{0\max}} \frac{R_{\mathrm{d5}} V_{\mathrm{d0}} F(R_0)}{|V_{\mathrm{d5}}(R_0, R_{\mathrm{d5}})|} \mathrm{d}R_0$$

[revised manuscript text omitted]

---

## Author Response (AR3)

Dear Dr. Andrew J. Kavanagh and Dr. Christoph Jacobi,

On behalf of my co-authors, we thank you very much for giving us the third opportunity to revise our manuscript entitled "The research on small-scale structures of ice particle density and electron density in the mesopause region" (#angeo-2019-10). We would like to express our great appreciation to you and the reviewer for these constructive comments and suggestions on our manuscript. Based on the comments and requests, we have made careful modification on the original manuscript. We attached the revised manuscript and responded every question from the reviewer.

Looking forward to hearing from you.

Thank you and best regards!

Yours sincerely

Ruihuan Tian

Dear Reviewer,

Thank you for your insightful comments concerning our manuscript entitled "The research on small-scale structures of ice particle density and electron density in the mesopause region". Those comments are all valuable and very helpful for revising and improving our paper, as well as the important guiding significance to our researches. We have studied these comments carefully and have made corrections which we hope make our paper more acceptable. The responds to the comments are as following. Once again, special thanks to you for good comments and hope that the correction will meet with approval.

**Responses to Reviewer**

The authors have made further useful revisions on the manuscript. However, there are three further important points the authors must carefully consider and address before the manuscript is suitable for publication.

First Point: There is still the question if the work is novel. As underscored (more clearly) by the authors in the current revision of the manuscript, the particle growth model is the primary novelty. However, the ultimate result of this work is to show if the electron density structures produced with the new particle growth model are in line with experiments and can provide further insight into the process producing PMSE. The results of the author's work conclude that meter scale electron irregularities are produced which is already well known and the question again arises if the work is novel. For instance, could other models produce meter scale irregularities?

Response

Thank you very much for pointing it out.

The main novelty of our work is to give a possible explanation on the formation of the ice particle density irregularities in the PMSE region. Through our model, we can obtain small-scale structures of ice particle density, which is consistent with the observations. As far as we know, no similar researches have been reported. Then the influence of the ice particle density structures on plasma is calculated only to show the corresponding meter scale electron irregularities and make the work more complete.

Lie-Svensen's model can also produce meter scale electron irregularities (Lie‑Svendsen, et al. 2003), but they use a prescribed ice particle density without explaining how such ice particle small-scale structures are produced. Kopnin et al. use dust acoustic solitons to explain the localized structures of the charged dust particles in the PMSE region (Kopnin, et al. 2004), unfortunately, the spatial scale of these structures is much smaller than the observed scale and much smaller than the wavelength of VHF radar. However, in our model, the meter scale ice particle density irregularities are produced successfully, which can further determine the formation of the corresponding meter scale electron irregularities, and we believe this is the advantage that other models don't have.

As noted in the second review (point 4), the authors need to consider if there are some parameter regimes that their current particle growth model can explain that other models cannot. A simple thing to do is to consider what parameters in the growth model controls the spatial scales of the irregularities. The authors should note that the work of Lie-Svensen considers a relatively broad range of spatial scales of irregularities from meters to 10's of meters (using a prescribed aerosol particle density). This is of course motivated by PMSE being observed at a range of frequencies (e.g. UHF, VHF, MF, and HF) and altitudes. An important question is what actually causes the variation in spatial scales in electron structures (and the associated aerosol structures calculated with the new growth model). Are the spatial

scales related to wind velocity or altitude for instance. Examples (and related work) are Bremer et al., PMSE observations at three different frequencies in northern Europe during summer, Annales Geophysicae, 1996 and Acala et al. Multifrequency observations of polar mesosphere summer echoes using alaskan radar facilities: Comparisons and scattering calculations, Radio Science, 2009.

To reiterate, the authors should consider which parameters in the new growth model can determine the spatial scales of irregularities (as noted in point 4. of the previous review). Is this the wind speed for instance? Also, what is the impact of altitude on the spatial scales of the irregularities? Can this be determined with the new growth model? Are longer or shorter irregularities produced as the altitude varies. These are all important questions that could potentially be answered with the model and be impactful to the community.

**Response**

Thank you very much for your valuable and instructive suggestions.

In the previous version of our manuscript, we have only focused on the model development and the influence of some environmental factors, such as vertical wind speed and altitude, has been ignored. While, the absence of discussion about environmental factor effects on the final calculation results may lead to a lack of persuasion of our model. So we have gotten some new calculation results with environmental factors (the vertical wind speed, altitude, the water vapor density and the size distribution of condensation nuclei) considered. The corresponding conclusion and discussion are presented as follow.

The spatial scale of the ice particle density irregularities is indeed affected by the vertical wind speed and altitude.

The spatial scale of the irregularities increases with the increase of wind speed, because lager wind speed corresponds to larger critical particle radius in the growth model and further leads to longer time scale and spatial scale of particle growth and movement.

The altitude is directly related to the neutral gas density. With the increase of

altitude, the neutral density becomes smaller, which leads to smaller critical particle radius and shorter spatial scale of the ice particle density irregularities.

In addition, the water vapor density can also influence the spatial scale of the irregularities. When the water vapor density increases, the spatial scale of the irregularities decreases, because a lager vapor density results in a lager change rate of particle radius. Then the particles can reach the inversion condition faster, and the reverse position is closer to the boundary, which means the spatial scale of the density structures gets shorter.

Finally, the size distribution of condensation nuclei will affect the spatial scale and the maximum value of the ice particle density structures obviously. The broader characteristic width of the radius distribution function corresponds to larger spatial scale and smaller maximum value of the density structures.

In our research, the electron density is anti-correlated to the density of charged ice particles, therefore the spatial scale of the electron density irregularities is also affected by these parameters: the vertical wind speed, altitude, the water vapor density and the characteristic width of the radius distribution function. It is remarkable that at lower altitude the spatial scale of the electron density irregularities is longer than that at higher altitude, which agrees with Bremer et al.'s view on explaining the phenomenon that, at lower altitude, the PMSE signals detected by long-wavelength radar (half wavelength = 54 m) are stronger than those detected by short-wavelength radar (half wavelength = 2.8 m) (Bremer, et al. 1997).

The influence of the condensation nucleus radius distribution function on the spatial scale of the irregularities has been shown in the first response (point 8), so it will not be included in the modified manuscript for the sake of simplicity.

The detailed analysis of other parameters' effect has been added in the Results and discussion section and the corresponding summary has been added in the Conclusion section.

Second Point: The authors use the term 'stochastic model' for the charging. Are the authors confident they are actually using a stochastic model? This would imply using a probability model to determine the discrete charging state. It may be more appropriate to describe this as a 'discrete charging model' (as opposed to continuous charging) and not a stochastic model. The Lie-Svensen model uses a discrete charging model not stochastic.

Response

Thank you very much for pointing it out. We have modified the description of the charging model in the Model section.

Third Point: There are still lots of issues with English language and typographical errors. The manuscript must be more carefully edited.

Response

Thank you very much for pointing it out. We have gone through the manuscript and revised some English usage and grammar mistakes to make it easier to understand.

Reference

[revised manuscript text omitted]

---

## Author Response (AR4)

Dear Dr. Andrew J. Kavanagh and Dr. Christoph Jacobi,

On behalf of my co-authors, we thank you very much for giving us the opportunity to revise our manuscript entitled "The research on small-scale structures of ice particle density and electron density in the mesopause region" (#angeo-2019-10). We have modified the grammatical problems in the original manuscript with the help of a commercial service in order to make it easier to understand.

Looking forward to hearing from you.

Thank you and best regards!

Yours sincerely

Ruihuan Tian

Dear Reviewer,

Thank you for pointing out the grammatical problem in our manuscript entitled "The research on small-scale structures of ice particle density and electron density in the mesopause region". This is very helpful for improving our paper. We have modified the grammatical problems in the original manuscript carefully to make our paper more acceptable.

[revised manuscript text omitted]
_{\rm d}\frac{{\rm d}\boldsymbol{u}_{\rm d}}{{\rm d}t}+(\boldsymbol{u}_{\rm d}-\boldsymbol{u})\frac{{\rm d}m_{\rm d}}{{\rm d}t}=m_{\rm d}\boldsymbol{g}-\mu_{\rm dn}m_{\rm d}(\boldsymbol{u}_{\rm d}-\boldsymbol{u})+q_{\rm d}\boldsymbol{E} \tag{1}$$

where $m_{\rm d}$, $\boldsymbol{u}_{\rm d}$, and $q_{\rm d}$ are the mass, velocity, and charge of ice particles, respectively. $\boldsymbol{u}$ is the velocity of neutral gas, $\boldsymbol{g}$ is gravitational acceleration, $\mu_{\rm dn}$ is the collision frequency between ice particles and gas, and $\boldsymbol{E}$ is the electric field. The electric force has a trivial effect on the motion of ice particles, because the charge-mass ratio of particles is usually very small (Jensen and Thomas 1988;Pfaff, et al. 2001). The inertial term is also negligible as its magnitude is much smaller than gravity (Garcia and Solomon 1985).

The water vapor is supersaturated in the polar mesopause region (Lübken 1999) and it is assumed that the size of condensation nuclei is larger than the condensation critical size, so stable growth of ice particles will continue when water molecules collide with particles during thermal motion. Ignoring reverse processes such as sublimation, the mass change rate of ice particles is

$$\frac{{\rm d}m_{\rm d}}{{\rm d}t}=\mu_{\rm wd}m_{\rm w} \tag{2}$$

The collision frequency between water vapor and ice particles is $\mu_{\rm wd}=n_{\rm w}\pi r_{\rm d}^{2}v_{\rm w}$ based on the hard-sphere collision model (Lieberman and Lichtenberg 2005), in which $m_{\rm w}$, $n_{\rm w}$ and $v_{\rm w}$ are the mass, number density, and thermal velocity of water molecules, respectively.

The collision frequency between air molecules and ice particles in the neutral drag force term is(Schunk 1977)

$$\mu_{\rm dn}=\frac{8}{3\sqrt{\pi}}\frac{n_{\rm n}m_{\rm n}}{m_{\rm d}+m_{\rm n}}\sqrt{\frac{2k_{\rm B}T_{\rm g}(m_{\rm d}+m_{\rm n})}{m_{\rm d}m_{\rm n}}}\pi(r_{\rm d}+r_{\rm n})^{2} \tag{3}$$

where $n_{\rm n}$, $m_{\rm n}$, and $r_{\rm n}$ are number density, mean molecule mass, and effective radius of neutral molecule, respectively, and $T_{\rm g}$ is the gas temperature. The neutral molecule mass $m_{\rm n}$ is assumed to be $28.96m_{\rm u}$, in which $m_{\rm u}$ is the proton mass.

According to Eq. (1) the velocity of ice particles is obtained as

$$\boldsymbol{u}_{\rm d}=\boldsymbol{u}+\frac{m_{\rm d}}{\mu_{\rm dn}m_{\rm d}+\mu_{\rm wd}m_{\rm w}}\boldsymbol{g} \tag{4}$$

On the basis that $n_{\rm w}<<n_{\rm n}$(Seele and Hartogh 1999), $m_{\rm w}<<m_{\rm d}$, $m_{\rm n}<<m_{\rm d}$, $r_{\rm n}<<r_{\rm d}$, $v_{\rm n}\sim v_{\rm w}$, and taking vertical up to be the positive direction, the velocity of ice particles is simplified as

$$u_{\rm d}=u-g/\mu_{\rm dn} \tag{5}$$

Ice particles are composed of condensation nuclei and the attached ice. The mass of

a single ice particle is

$$m_d = \frac{4}{3}\pi r_0^3 \rho_0 + \frac{4}{3}\pi (r_d^3 - r_0^3)\rho_d \tag{6}$$

where $r_0$ and $\rho_0$ are the initial radius and mass density of condensation nuclei, and $\rho_d$ is the mass density of ice.

Based on the expressions of $m_d$ and $\mu_{dn}$, the relation between ice particle velocity and radius is

$$u_d = u - \frac{g}{n_n m_n v_n}[\rho_d r_d + (\rho_0 - \rho_d)\frac{r_0^3}{r_d^2}] \tag{7}$$

At the boundaries of the condensation region $r_d = r_0$, and the initial velocity of condensation nuclei is

$$u_{d0} = u(1 - r_0/r_c) \tag{8}$$

where $r_c$ is the critical radius:

$$r_c = n_n m_n v_n u/(g\rho_0) \tag{9}$$

When the radius of condensation nuclei $r_0 > r_c$, gravity is larger than the neutral drag force, $v_{d0} < 0$, and particles move downwards. Otherwise, particles move upwards.

Based on the relation between $m_d$ and $r_d$, the change rate of ice particle radius is

$$\frac{dr_d}{dt} = \frac{1}{4}\frac{n_w m_w v_w}{\rho_d} = c \tag{10}$$

It can be clearly observed that the ice particle radius increases linearly with time:

$$r_d = r_0 + ct \tag{11}$$

The particle trajectory can then be obtained by the following integral

$$z - z_0 = \int_0^t u_d dt = c^{-1}\int_{r_0}^{r_d} u_d dr_d \tag{12}$$

where $z_0$ is the reference height where condensation nuclei enter the studied region. In this work $z_0 = 0$ is set at the lower boundary and $z_0 = h$ is set at the upper boundary, where $h$ is the distance between the two boundaries.

It is assumed that the condensation nucleus radius ranging from $r_{0min}$ to $r_{0max}$ has a certain distribution function $f(r_0)$. The density of condensation nuclei with radius in the small range $r_0 \to r_0 + dr_0$ is $dn(r_0) = f(r_0)dr_0$, and their velocity is $u_{d0}$. When these particles arrive at height $z$, their radius increases to $r_d(r_0, z)$, the corresponding number density turns into $dn(r_0, z)$, and the velocity becomes $u_d(r_0, z) = u_d[r_0, r_d(r_0, z)]$. According to the particle-conservation law

$$u_{d0}dn(r_0) = u_d(r_0, z)dn(r_0, z) \tag{13}$$

The number density of ice particles at height $z$ can then be obtained by

$$n_{\rm d}(z) = \int {\rm d}n(r_0, z) = \int_{r_{0\rm min}}^{r_{0\rm max}} \frac{u_{\rm d0} f(r_0)}{u_{\rm d}(r_0, z)} {\rm d}r_0 \tag{14}$$

The averaged ice particle radius at height $z$ is

$$\overline{r_{\rm d}}(z) \;\; = \frac{\int r_{\rm d}(z) {\rm d}n(r_0, z)}{n_{\rm d}(z)} \tag{15}$$

By integrating all condensation nucleus radii, a stable distribution of $n_{\rm d}$ and $r_{\rm d}$ can be obtained. The particles continue to enter and leave the condensation region, and as long as the external environment does not change, the distribution of particle density and radius will remain unchanged. The influence of these stable $n_{\rm d}$ and $r_{\rm d}$ profiles on electron and ion density is then calculated.

Considering ionization, electron-ion recombination, and ion loss on ice particles, the continuity equation of ion density can be written as

$$\frac{\partial n_{\rm i}}{\partial t} + \frac{\partial(n_{\rm i} u_{\rm i})}{\partial z} = Q - \alpha n_{\rm i} n_{\rm e} - D^+ n_{\rm i} \tag{16}$$

Ignoring gravity, the drift velocity of ions $u_{\rm i}$ is determined by

$$u_{\rm i} = \frac{eE}{m_{\rm i} \mu_{\rm in}} - \frac{k_{\rm B} T_{\rm g}}{m_{\rm i} \mu_{\rm in}} \frac{1}{n_{\rm i}} \frac{\partial n_{\rm i}}{\partial z} \tag{17}$$

The electric field $E$ is predominantly determined by electron density gradient because the diffusion coefficient and mobility of electrons are much larger than that of ions:

$$E = -\frac{k_{\rm B} T_{\rm g}}{e} \frac{1}{n_{\rm e}} \frac{\partial n_{\rm e}}{\partial z} \tag{18}$$

In the typical PMSE layer, there are several kinds of ions carrying one unit positive charge: $N_2^+$, $O_2^+$, $NO^+$, and $H^+(H_2O)_n$. As specified by Reid (Reid 1990), the averaged ion parameters $n_{\rm i}$, $m_{\rm i}$, and $T_{\rm g}$ are applied to describe the density, mass, and temperature of ions, respectively, and the averaged ion mass $m_{\rm i}$ is set as $50m_{\rm u}$ at 85 km altitude. According to Hill and Bowhill's theory (Hill and Bowhill 1977), the ion-neutral collision frequency is

$$\mu_{\rm in} = 2.6 \times 10^{-15} n_{\rm n} \left( 0.78 \frac{28}{M_{\rm i}+28} \sqrt{1.74 \frac{M_{\rm i}+28}{28 M_{\rm i}}} \right.$$
$$\left. + 0.21 \frac{32}{M_{\rm i}+32} \sqrt{1.57 \frac{M_{\rm i}+32}{32 M_{\rm i}}} + 0.01 \frac{40}{M_{\rm i}+40} \sqrt{1.64 \frac{M_{\rm i}+40}{40 M_{\rm i}}} \right) \tag{19}$$

[revised manuscript text omitted]